# Tumor suppression in basal keratinocytes via dual non-cell-autonomous functions of a Na,K-ATPase beta subunit

**Julia Hatzold[1,2]\*, Filippo Beleggia[2,3,4], Hannah Herzig[5], Janine Altmüller[3,6], Peter Nürnberg[2,4,6], Wilhelm Bloch[5], Bernd Wollnik[2,3,4,7], Matthias Hammerschmidt[1,2,4]\***

[1]Institute for Zoology, Developmental Biology Unit, University of Cologne, Cologne, Germany; [2]Center for Molecular Medicine Cologne, University of Cologne, Cologne, Germany; [3]Institute of Human Genetics, University Hospital Cologne, Cologne, Germany; [4]Cologne Excellence Cluster on Cellular Stress Responses in Aging-Associated Diseases, University of Cologne, Cologne, Germany; [5]Institute of Cardiology and Sports Medicine, German Sport University Cologne, Cologne, Germany; [6]Cologne Center for Genomics, University of Cologne, Cologne, Germany; [7]Institute of Human Genetics, University Medical Center Göttingen, Göttingen, Germany

**\*For correspondence:** jhatzold@uni-koeln.de (JH); mhammers@uni-koeln.de (MH)

**Competing interests:** The authors declare that no competing interests exist.

**Abstract** The molecular pathways underlying tumor suppression are incompletely understood. Here, we identify cooperative non-cell-autonomous functions of a single gene that together provide a novel mechanism of tumor suppression in basal keratinocytes of zebrafish embryos. A loss-of-function mutation in *atp1b1a*, encoding the beta subunit of a Na,K-ATPase pump, causes edema and epidermal malignancy. Strikingly, basal cell carcinogenesis only occurs when Atp1b1a function is compromised in both the overlying periderm (resulting in compromised epithelial polarity and adhesiveness) and in kidney and heart (resulting in hypotonic stress). Blockade of the ensuing PI3K-AKT-mTORC1-NFκB-MMP9 pathway activation in basal cells, as well as systemic isotonicity, prevents malignant transformation. Our results identify hypotonic stress as a (previously unrecognized) contributor to tumor development and establish a novel paradigm of tumor suppression.

## Introduction

Many malignancies result from loss-of-function mutations in one or more tumor suppressor genes whose normal function is concerned with the inhibition of cell division, the induction of apoptosis and/or the inhibition of metastasis. Most tumor suppressors affect one or more of these processes in a cell-autonomous manner, being produced by and acting within the tumor precursor cells themselves (*Sherr, 2004*; *Sun and Yang, 2010*), whereas comparably few genes are known to block tumorigenesis in a non-cell-autonomous manner (*Chua et al., 2014*). Tumor suppressors often act by inhibiting or antagonizing proto-oncogenic factors. The phosphatase PTEN for instance is the antagonist of phosphatidyl-inositol-3-kinases (PI3Ks), critical coordinators of intracellular signaling in response to extracellular stimuli such as growth factors and cytokines. Hyperactivity of PI3K signaling cascades, including that involving AKT/PKB (protein kinase B), is one of the most common events in human cancers (*Altomare and Testa, 2005*; *Thorpe et al., 2015*).

One of the transcription factors regulated by PI3K/AKT via the mTORC1 complex is NFκB (*Dan et al., 2008*). Increased NFκB activity is observed in many carcinomas (solid malignancies

**eLife digest** Cancer can develop when cells in the body gain mutations that allow them to grow and divide rapidly. Some of these mutations may affect the activity of genes that usually act to prevent cancer from developing. Several such "tumor suppressor" genes have been identified, but it is likely that many remain undiscovered and it is far from fully understoodhow all these genes work. One way to identify new tumor suppressor genes is to examine tumors to search for genes that have gained mutations that block their activity, known as loss-of-function mutations.

Hatzold et al. identified a new and rather unexpected tumor suppressor gene by studying a zebrafish mutant that develops skin cancer as the embryo grows. The experiments showed that cells in the skin of the developing embryos of this mutant grow excessively and start to invade deeper tissues in the body. This behavior is caused by loss-of-function mutations in a gene called *atp1b1a*. This gene encodes part of an ion pump protein that helps to control the amount of water and ions in cells and in body fluids.

Further experiments showed that this tumor suppressor gene does not act in the skin cells themselves but in other cells of organs such as the kidney. The kidney is involved in controlling the water and ion content of the body (known as osmoregulation), and the *atp1b1a* mutants have lower levels of ions and increased levels of water than normal zebrafish. Cancer formation could be completely blocked when the mutant embryos were kept in a solution that had the same salt and water content as the animals, instead of regular fresh water. This suggests that exposure of cells to body fluids with decreased ion and increased salt contents, a condition also called hypotonic stress, increases the risk of developing some tumors.

Osmoregulatory organs that are not working efficiently, or injuries that expose cells to different ion and water levels can both cause hypotonic stress. The next steps are to investigate whether this stress also promotes cancer formation in mammals, including humans.

derived from epithelial cells), promoting cell survival, proliferation and metastasis (*Karin et al., 2002*). However, the actual genetic cause of NFκB activation is unknown in most of these cases (*Ben-Neriah and Karin, 2011*). Furthermore, in other instances, anti-tumorigenic effects of NFκB have been described (*Ben-Neriah and Karin, 2011*).

In epithelial cells, carcinogenesis can also be caused by compromised functioning of genes involved in the formation and maintenance of epithelial cell polarity (*Martin-Belmonte and Perez-Moreno, 2012*; *Ellenbroek et al., 2012*). Prominent examples of affected proteins include CRB3, a member of the Crumbs complex; PAR3, a member of the partitioning defective (PAR) / aPKC complex, which like the Crumbs complex promotes apical identities; and LGL1 (Lethal giant larvae-1), DLG (Discs large) and SCRIB, members of the Scribble complex, which promote basolateral identities. In *Drosophila*, a second pro-basolateral complex has been described, consisting of a Na,K-ATPase, Coracle, Yurt and Neurexin IV, which acts in partial functional redundancy with the Scribble complex (*Paul et al., 2007*; *Laprise et al., 2009*).

Na,K-ATPases are ion pumps composed of a catalytic α-subunit and a regulatory β-subunit that is required for proper trafficking, localization, and functionality of the α-subunit (*Geering, 2008*). Four α-subunit and three β-subunit genes have been described in mammals, of which α1 and β1 are the major isoforms in epithelial cells. These subunits transport Na and K ions across the cell membrane, and thereby play well-characterized roles in generating electrochemical gradients in multiple cell types, and in sodium and water balancing in renal tubules, thereby regulating body fluid composition and volume. However, additional pump-independent and evolutionary conserved functions for Na,K-ATPase proteins have been described in the context of epithelial cell adhesiveness and polarity (*Vagin et al., 2012*). Here, the β-subunits play a prominent role. These are type II transmembrane proteins with glycosylated extracellular domains that can form β1-β1 trans-bonds between neighboring cells, thereby promoting both epithelial polarity and intercellular adhesiveness of cultured cells. β-subunit glycosylation has a positive impact on cell-cell adhesiveness, most likely by stabilizing E--cadherin cell adhesion proteins (*Vagin et al., 2008*). Consistently, in transformed Madin-Darby canine kidney epithelial cells (MDCK) cells, the expression of both E-cadherin and the Na,K-ATPase

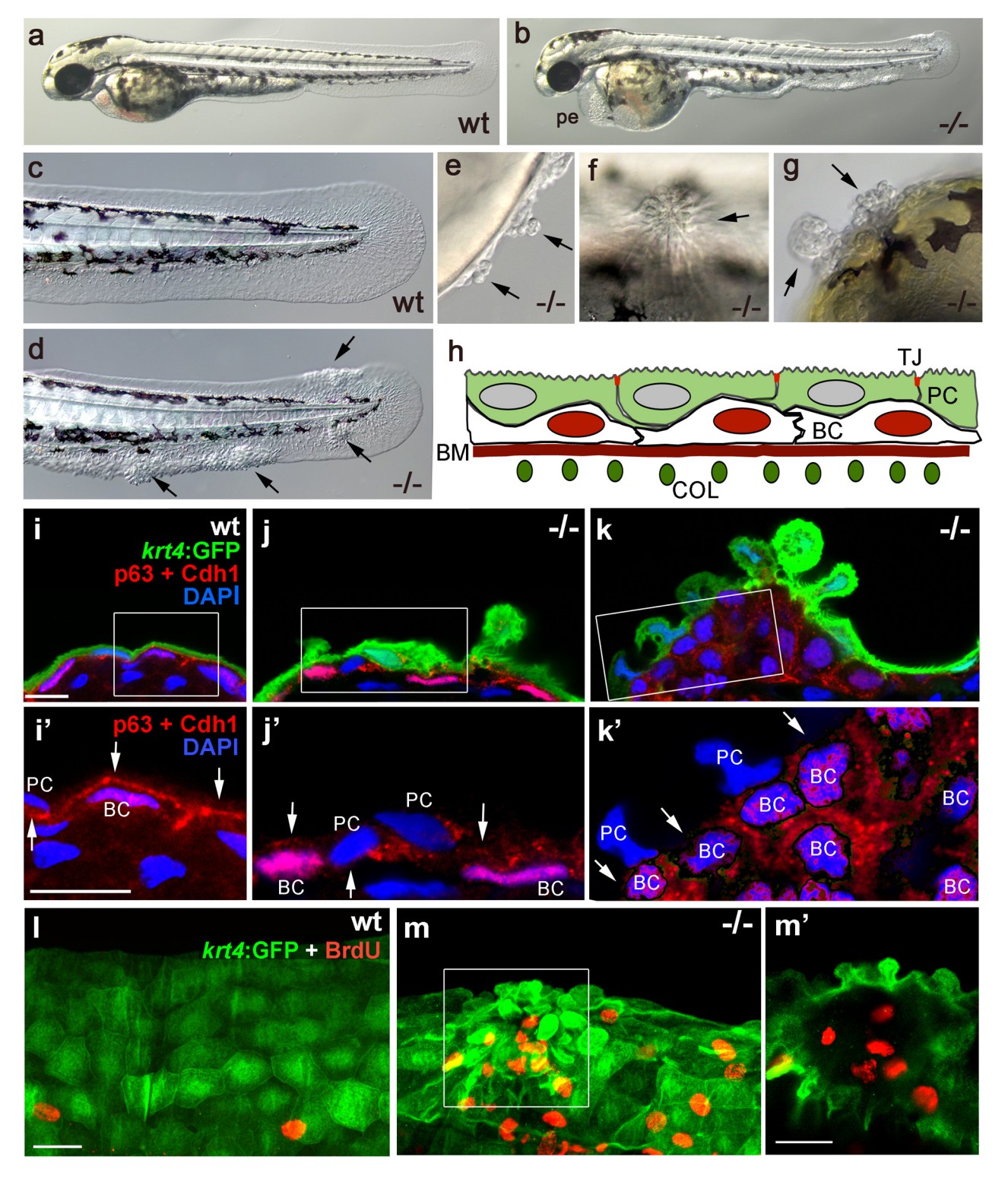

**Figure 1.** Epidermal aggregates in *psoriasis* mutants display hyperplasia. (a–g) Live images of wt siblings (a,c) and *psoriasis* mutants (b,d–g); *psoriasis* mutants develop pericardial edema (pe) and epidermal aggregates on the medium fin fold (b, d), on the yolk sac (e), on the flank (f) (all at 54 hpf), and on the head (g; 72 hpf). (h) Schematic of embryonic skin. Peridermal cells (PC, green) with apical tight junctions (TJ, red) are located above p63 basal keratinocytes (BC, red nuclei). The basement membrane (BM, red) separates the basal epidermal layer from the dermis containing collagen fibers (COL, green). (i–k) IF of periderm-specific GFP (green), Cdh1 (red), and p63 (red) on transverse sections through 48 hpf *Tg(krt4:GFP)* wt and *psoriasis* mutant

*Figure 1 continued on next page*

*Figure 1 continued*

embryos, counterstained with DAPI (blue). (i'–k') show magnified views of regions framed in (i–k), without the green channel. In wt, the epidermis is bi-layered, with flat cells (i) and Cdh1 is localized at cell borders between peridermal and basal cells (i'; arrows). In an early-stage aggregate of the mutant, peridermal cells have rounded up (j), and Cdh1 levels are reduced at cell borders (j',k'; arrows). An advanced aggregate (k) contains multi-layered basal epidermal cells (k'). Scale bar: 10 µm. (l,m) Whole mount IF of incorporated BrdU (red) and periderm-specific GFP (green) in 54 hpf *Tg(krt4:GFP)* wt sibling (l) and *psoriasis* mutant (m), showing elevated numbers of BrdU-positive non-peridermal cells in aggregates. (m) Maximum intensity projection of a confocal Z stack through aggregate; (m') single focal plane. Scale bars: 20 µm. Abbreviations: BC, basal cell; BM, basement membrane; COL, collagen fibers; PC, peridermal cell; pe, pericardial edema; wt, wild-type.

β1-subunit is drastically reduced, and epithelial polarity and junctional complexes can only be re-established upon repletion of both molecules (*Rajasekaran et al., 2001*). Na,K-ATPase $\beta_1$-subunit levels are also reduced in cell lines derived from various carcinomas. This evidence, together with functional studies with transformed MDCK cells, has led to the proposal that the Na,K-ATPase $\beta_1$ subunit has a potential tumor-suppressor function (*Inge et al., 2008*), but direct genetic evidence for this notion had been missing to date.

Here, we report that a loss-of-function mutation in the β-subunit Atp1b1a causes epidermal malignancy in zebrafish *psoriasis* mutant embryos (*Webb et al., 2008*), correlated with reduced E--cadherin levels and a genetic interaction with the formerly described epithelial polarity regulator and tumor suppressor Lgl2 (*Sonawane et al., 2005*; *Reischauer et al., 2009*). During the affected stages, the epidermis is normally bi-layered, consisting of a tight junction-bearing outer periderm and a basal layer of keratinocytes. The latter are transformed in *atp1b1a* mutants, leading to their overgrowth and invasion of dermal compartments. Keratinocyte transformation is transduced via aberrant activation of a PI3K-AKT-mTORC1-NFκB-MMP9 (metalloprotease 9) pathway. Chemical inhibition of PI3K, mTORC1 and NFκB rescues all aspects of malignancy, whereas knockdown of MMP9 alleviates only epidermal invasiveness but not hyperplasia, pointing to a specific role of this matrix metalloprotease as one of the mediators of metastasis, and to the involvement of additional relevant NFκB targets. Epidermal malignancy is also fully suppressed upon incubation of embryos in isotonic (rather than the natural hypotonic) medium, and the remaining basal cell polarity and adhesiveness defects can be rescued by concomitant re-introduction of wild-type Atp1b1a in peridermal cells. Together with other presented data, these findings indicate that epidermal malignancy results from a combined loss of the β-subunit's trans-layer function in promoting basal cell polarity via the periderm, and its osmoregulatory function in suppressing hypotonic stress. Possible tumor-promoting effects of hypotonicity during human carcinogenesis are discussed.

## Results

### *psoriasis* mutant embryos display characteristics of epidermal malignancy

The zebrafish *psoriasis* mutant was isolated in a phenotype-based screen after undirected ethyl methanesulfonate (EMS)-mutagenesis and has been described as developing edema as well as epidermal aggregates during embryogenesis (*Figure 1a,b*) (*Webb et al., 2008*). Aggregates were preferentially found on the median fin folds, but also on the flanks, the yolk sac and the head (*Figure 1c–g*). Analysis with specific molecular markers revealed that the aggregates consist of both peridermal cells and basal keratinocytes (see *Figure 1h* for schematic of embryonic skin), and can already be detected at 48 hours post-fertilization (hpf) (*Figure 1i–k*). Initial defects were more pronounced in the outer periderm, in which the normally flat cells started to round up (*Figure 1i,j*). In addition, cell membranes, in particular in basal domains facing the underlying basal keratinocytes, displayed reduced levels of the cell-cell adhesion molecule E-cadherin (Cdh1) (*Figure 1i',j'*). Altered organization of basal keratinocytes themselves was only obvious in more advanced aggregates, in which cells had lost their regular epithelial shape and their mono-layered organization (*Figure 1k*). At 56 hpf, hyper-proliferation was evident in basal cells, whereas peridermal cells did not proliferate (*Figure 1l,m*).

Live time-lapse imaging revealed partial epithelial-mesenchymal transitions (EMTs) in the basal keratinocytes of mutant embryos at 48 hpf, characterized by the formation of cellular processes and

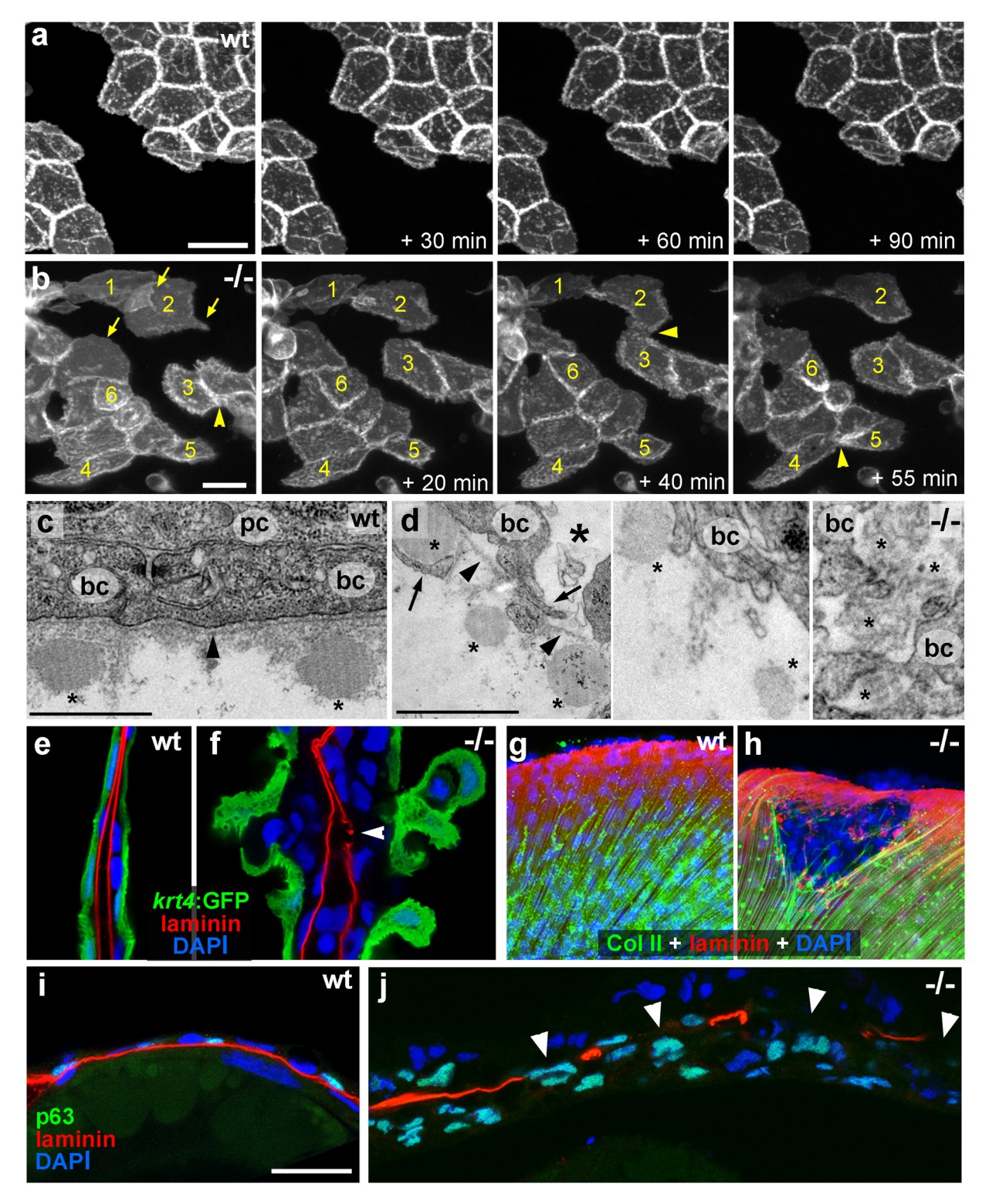

**Figure 2.** *psoriasis* keratinocytes display partial EMT and invasive behavior. (a,b) Stills from in vivo time-lapse recordings (*Videos 1* and *2*) of clones of membrane-bound GFP-labelled (*Tg(Ola.Actb:Hsa.hras-egfp)*) basal keratinocytes in a mosaic wt (a) or an *atp1b1a* morphant (b) embryo; 'n min' indicates elapsed time since the start of the recordings at 48 hpf (*Videos 1* and *2*). Wild-type cells form a rigid epithelium and maintain their shapes and relative positions (a), whereas *atp1b1a* morphant cells form cellular processes (arrows), dynamically dis- and re-associate (arrowheads) and eventually crawl on top of each other (cell 1; b, first panel). Cell 1 moves out of the focal plane after 60 min, cell 6 changes its shape from roundish to
*Figure 2 continued on next page*

*Figure 2 continued*

more hexagonal and vice versa. Scale bars: 20 µm. (**c,d**) Transverse TEM sections through median fin fold, at 58 hpf. In wt (**c**), an intact basement membrane (BM; black arrowhead) separates the compact layer of basal cells from the underlying dermis, which contains actinotrichia (small asterisks). The *psoriasis* mutant (**d**) displays large intercellular gaps (large asterisk) between basal cells, cellular protrusions (arrows) of basal cells, a discontinued BM (arrowheads to remaining BM), direct contacts between epidermal cells, and disassembling dermal actinotrichia (small asterisks) that lose their regular shape and striated pattern. bc: basal cells; pc: peridermal cell. Scale bars: 1 µm. (**e,f**) Laminin and peridermal-specific GFP double IF, counterstained with DAPI, at 58 hpf. Transverse sections through the fin fold of *Tg(krt4:GFP)* transgenics reveal basement membrane fragmentation (arrowhead) below an epidermal aggregate in the mutant (**f**). (**g,h**) Laminin and type II collagen double IF, counterstained with DAPI at 58 hpf; view of fin folds of whole mounts, showing basement membrane fragmentation and actinotrichia disassembly in the mutant (**h**). (**i,j**) Laminin and p63 double IF, counterstained with DAPI at 58 hpf; transverse section through the yolk sac. Arrowheads in (**j**) point to holes in the basement membrane of the mutant. Note the presence of p63 keratinocytes below the basement membrane in the dermal space. Scale bar: 20 µm.

The following source data and figure supplements are available for figure 2:

**Figure supplement 1.** *psoriasis* mutants display local degradations of the skin basement membrane and of underlying collagenous actinotrichia of the dermis .

**Figure supplement 2.** *psoriasis* mutants display skin inflammation, which does not contribute to the formation of epidermal aggregates.

**Figure supplement 2–source data 1.** Source data for *Figure 2—figure supplement 2e*.

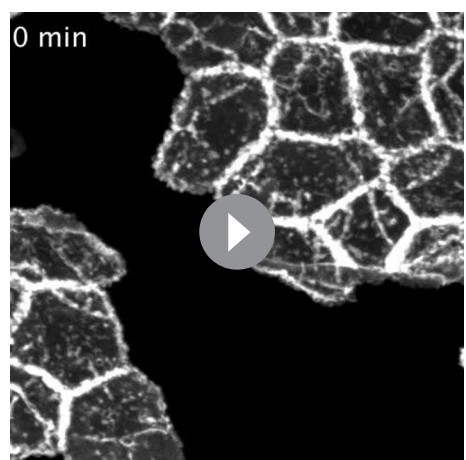

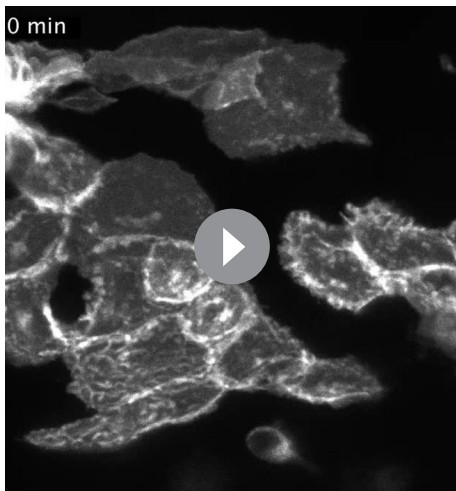

**Video 1.** In vivo time-lapse recordings revealing stable epithelial integrity of basal keratinocytes in wild-type embryo. In vivo time-lapse recordings of mosaic wt embryo with clones of basal keratinocytes labeled with membrane-bound GFP. The *Tg(Ola.Actb:Hsa.hras-egfp)*-bearing progenitors of these clones had been transplanted into a non-labeled host at 6 hpf. At 48 hpf, the mosaic embryo was mounted in 1.5% low-melting agarose in E3 medium and z-stacks of clusters of GFP-positive cells were recorded every five minutes with a Zeiss laser scanning microscope (Zeiss LSM710 META) for 95 min. Maximum intensity projections were processed using ImageJ software. Stills of the Videos are shown in *Figure 2a*. Similar results were obtained for nine recordings from 9 different individuals.

**Video 2.** In vivo time-lapse recordings revealing partial EMT of basal keratinocytes in *atp1b1a* morphant embryo. In vivo time-lapse recordings of mosaic *atp1b1a* morphant embryo with clones of basal keratinocytes labeled with membrane-bound GFP. The *Tg(Ola.Actb:Hsa.hras-egfp)*-bearing progenitors of these clones had been transplanted into a non-labeled host at 6 hpf. At 48 hpf, the mosaic embryo was mounted in 1.5% low-melting agarose in E3 medium and z-stacks of clusters of GFP-positive cells were recorded every five minutes with a Zeiss laser scanning microscope (Zeiss LSM710 META) for 95 min. Maximum intensity projections were processed using ImageJ software. Stills of the videos are shown in *Figure 2b*. Similar results were obtained for nine recordings from 9 different individuals.

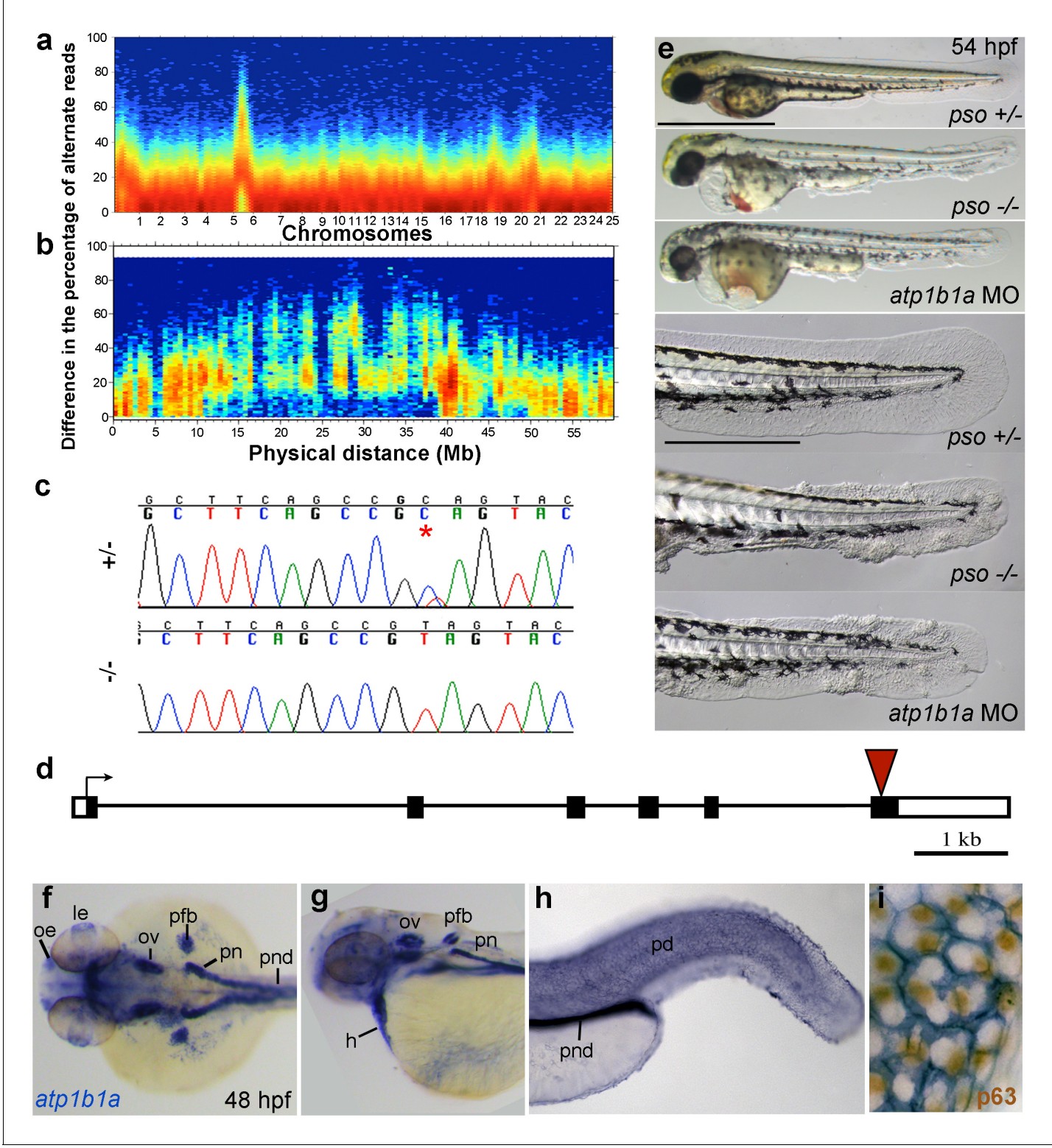

**Figure 3.** The *psoriasis* phenotype is caused by a loss-of-function mutation in *atp1b1a*, which is expressed in multiple epithelia, but not in basal keratinocytes. (a-d) Exome sequencing links the *psoriasis* mutation to LG6 and identifies a C to T transition in *atp1b1a*. (a,b) Heat maps showing the density of variant loci over the whole genome (a) and on chromosome 6 (b). The Y-axis shows the absolute value of the difference in the percentage of DNA harboring the variation between the pool of affected offspring and the pool of their parents. Most of the genome shows a difference close to 0%, indicating that the parental DNA segregated randomly in the affected offspring. Only one peak on chromosome 6 shows low density at a difference of

*Figure 3 continued on next page*

*Figure 3 continued*

0%, but high density at between 25% and 50% difference, which is expected at the linked locus under the assumption of a recessive mode of inheritance. (c) Chromatographs of Sanger sequencing of sibling and mutant DNA showing the *psoriasis* mutation (*). (d) Schematic representation of the *atp1b1a* locus. Red arrowhead indicates the position of the mutation. (e) Live images of 54 hpf wt siblings, *psoriasis* mutants, and *atp1b1a* morphants. MO-based knockdown of *atp1b1a* in wt embryos phenocopies both the pericardial edema and epidermal aggregates. Scale bars: 500 μm; 250 μm (magnifications). (f–i) WISH detects *atp1b1a* RNA (blue) in heart and multiple epithelia of 48 hpf wt embryos, including the pronephric duct and the periderm but not in the basal keratinocytes (f–h), as seen at higher magnification after counterstaining of nuclei of basal keratinocytes for p63 protein (i). The *atp1b1a* RNA signal is not detected around p63 nuclei, but is detected in hexagonal peridermal cells with p63⁻ nuclei. Abbreviations: h, heart; le, lense; oe, olfactory epithelium; ov, otic vesicle; pd, periderm; pfb, pectoral fin bud; pn, pronephros; pnd, pronephric duct.

The following source data and figure supplements are available for figure 3:

**Figure supplement 1.** Schematic of the genomic region between 27,899,072–30,685,841 on Chromosome 6 of Ensembl *Danio rerio* version 84.10 (GRCz10).

**Figure supplement 1—source data 1.** List of annotated genes, in syntenic order, contained in the genomic 2.76 Mb region shown in *Figure 3—figure supplement 1*, together with their chromosomal location, and their sequencing status.

dynamic dissociations and re-associations of cells (*Figure 2a,b* and *Videos 1,2*). Transmission electron microscopy (TEM) and immunofluorescence (IF) analyses of the median fin folds of *psoriasis* mutants revealed that at 58 hpf basal keratinocytes had loosened their intercellular connections (*Figure 2c,d*). In addition, the basement membrane (BM) underneath the basal layer was disintegrated to variable degrees (*Figure 2e–j*; *Figure 2—figure supplement1*). Furthermore, numerous basal cells had entered the dermal compartment (*Figure 2i,j*) where they were in close contact with the remnants of the actinotrichia, dermal collagenous structures that were significantly disassembled in regions with epidermal aggregates (*Figure 2c,d,g,h*; *Figure 2—figure supplement 1a–d*). Together, this evidence demonstrates that basal keratinocytes of *psoriasis* mutants display both hyperplasia and invasive behavior, characteristics of epidermal malignancy.

In addition, *psoriasis* mutants displayed moderate skin inflammation, characterized by increased numbers of *mpx*-positive neutrophils and *lyz*-positive macrophages at 54 hpf (*Figure 2—figure supplement 2a,c*; and data not shown). This inflammation does not, however, seem to contribute to epidermal malignancy, as epidermal aggregates of unaltered numbers and sizes were also obtained in *psoriasis* mutants after ablation of the myeloid lineage by *pu.1* antisense morpholino oligonucleotide (MO) injection (*Carney et al., 2007*) (*Figure 2—figure supplement 2a–e*).

## The *psoriasis* phenotype is caused by a loss-of-function mutation in *atp1b1a*, which encodes a Na,K-ATPase β-subunit

Applying meiotic mapping, Webb et al. had placed the *psoriasis* mutation within a defined region on LG 6, but the causing lesion had not been identified (*Webb et al., 2008*). We conducted whole exome sequencing of pooled mutant siblings and their (heterozygous) parents and confirmed the linkage to LG 6 (*Figure 3a,b*). In addition, we identified a C to T transition in exon 6 of the gene *atp1b1a*, which was confirmed by Sanger sequencing (*Figure 3c*). By contrast, exons of all other annotated genes in this region (from 1.32 Mb / 22 genes North to 1.43 Mb / 20 genes South of *atp1b1a*) contained no nonsense, non-conservative missense or splice site mutations (*Figure 3—figure supplement 1*).

*atp1b1a* encodes a β-subunit of a Na,K-ATPase, an ion pump that drives the directional transport of Na and K ions across cell membranes. The functional pump consists of two subunits, the catalytically active α-subunit and the regulatory β-subunit, which is a type II transmembrane protein. The identified C to T transition (C760T) results in the generation of a premature stop codon (Q254*) (*Figure 3d*), removing the highly conserved C-terminus of the β-subunit's extracellular domain. Upon knocking down *atp1b1a* in wild-type embryos with a previously described MO blocking *atp1b1a* translation (*Blasiole et al., 2006*), we obtained edema and epidermal phenotypes indistinguishable from those of the *psoriasis* mutants (*Figure 3e*). This indicates that the identified mutation is indeed the causative lesion and has a loss-of-function effect.

To gain first insights into potential sites of essential Atp1b1a functions, we determined the β-subunit's expression pattern in 24– 48 hpf wild-type embryos via whole mount in situ hybridization

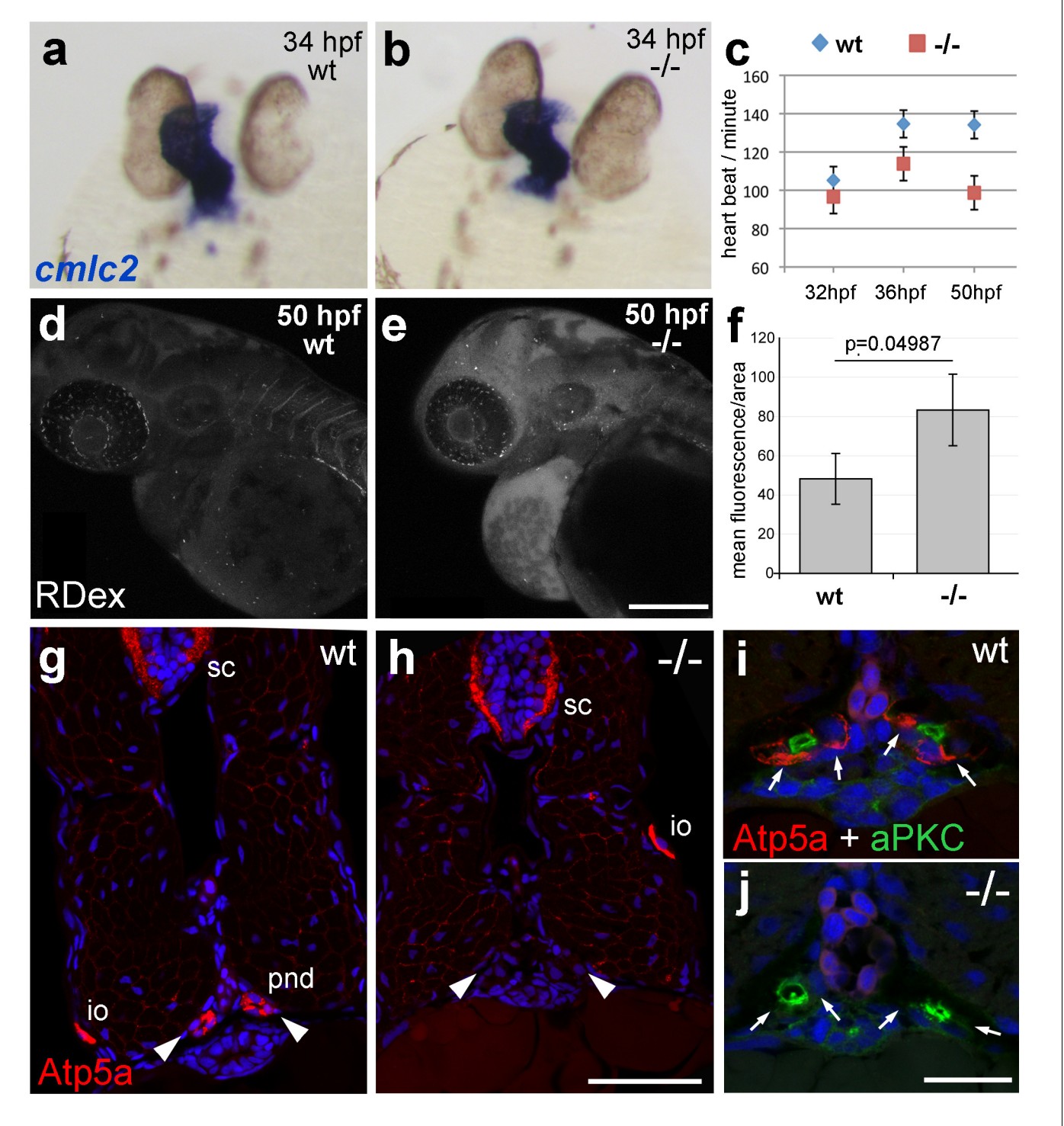

**Figure 4.** *atp1b1a* is required for proper heart and pronephric function. (**a**, **b**) WISH of *cmlc2* in 34 hpf embryos reveals normal heart tube elongation in *psoriasis* mutants. (**c**) *psoriasis* mutants exhibit a reduction of the heart beat. n = 12 (mutants), 24 (siblings). p values: 32 hpf: 3.8E-04, 36 hpf: 6.8E-07, 50 hpf: 1.2E-11. (**d**– **f**) *psoriasis* mutants show compromised clearance / excretion of rhodamine-dextrane injected into the cardinal vein at 34 hpf; confocal images of live embryos of wt sibling (**e**) and mutant (**f**) embryos at 50 hpf; RDex, rhodamine-dextrane; scale bar: 200 μm. (**f**) Quantification of mean fluorescence intensity of a defined area in confocal images, determined with ImageJ software; n = 3 for each condition. Error bars represent standard deviations. (**g–j**) IF of Atp5a (red) and aPKC (green), counterstained with DAPI (blue), on transverse sections of 54 hpf wt (**g,i**) and *psoriasis* mutant (**h,j**) embryos. Atp5a signal is absent from the pronephric duct (pnd, arrowheads) but not from ionocytes (io) or spinal cord (sc) in *psoriasis* mutants (**g,h**).

*Figure 4 continued on next page*

*Figure 4 continued*

The apical marker aPKC is still present in the pronephric duct cells of *psoriasis* mutants, outlining the lumen of the ducts, whereas Atp5a is missing from the basolateral site (arrows; **i,j**).

The following source data is available for figure 4:

**Source data 1.** Source data for *Figure 4*.

(WISH). Consistent with former reports (*Thisse et al., 2001*), *atp1b1a* was strongly expressed in the pronephros, the heart, and multiple (other) epithelia (*Figure 3f,g*). In addition, we found previously unrecognized expression in peridermal cells, but not in the basal keratinocytes of the epidermis (*Figure 3h,i*).

## *atp1b1a* is required for proper heart and pronephric function

The zebrafish Na,K-ATPase α-subunit Atp1a1a.1 has been formerly described as required for heart tube elongation and proper heart-beating (*Cheng et al., 2003*). *psoriasis* mutants also displayed a mildly reduced heartbeat rate at 33 hpf, which was more pronounced at 50 hpf (*Figure 4c*), but no defects in heart tube elongation were apparent at 34 hpf (*Figure 4a,b*). This suggests that *atp1b1a* is required for proper functioning of the embryonic heart rather than for its development, although the cellular basis of the heart malfunction in *psoriasis* mutants remains unclear. We could, however, identify specific defects in the organization of epithelial cells in the pronephros and its ducts. Antibodies raised against the chicken Na,K-ATPase α-subunits α5 and α6F have been reported to detect zebrafish α-isoforms in the embryonic kidney (*Drummond et al., 1998*) and in ionocytes (*Lin et al., 2006*) (but not in the heart and periderm; own data not shown). α5/α6F immunolabelling was completely absent in the pronephric duct of *psoriasis* mutants, but not in their ionocytes (*Figure 4g–j*), demonstrating that the mutation in the β-subunit results in a complete failure to target these α-subunits to the basolateral membrane of pronephric cells. A comparable mislocalization of α-subunits in the zebrafish pronephros has been associated with compromised osmoregulatory function and edema formation in several other instances (*Drummond et al., 1998*; *Hentschel et al., 2005*; *Martin-Belmonte and Perez-Moreno, 2012*).

Together, this evidence points to an essential role for Atp1b1a in the proper positioning of the Na/K pump in the basolateral membrane domain of kidney epithelial cells, which is crucial for the kidney's osmoregulatory function. Together with the reduced heart beat, these kidney defects should lead to compromised renal water secretion. Indeed, clearance of rhodamin-dextrane injected into the cardinal vein was significantly reduced in *psoriasis* mutants at 24 hr post injection (*Figure 4d–f*).

## Epidermal malignancy is dependent on hypotonic conditions

The osmoregulatory demands of an animal depend on its environmental conditions. Zebrafish live in freshwater and are estimated to have an internal osmolarity of 230–300 mOsm, whereas freshwater has 10 mOsms (*Enyedi et al., 2013*). As a consequence of this external hypotonicity, the embryos face a continuous passive influx of water that needs to be actively excreted via the pronephros. Therefore, compromised function of Atp1b1a in pronephros and heart causes edema formation as a result of increased water content, which is indicative of hypotonicity in interstitual compartments.

To bypass the need for Atp1b1a for osmoregulation, we incubated *psoriasis* mutant embryos in E3 medium containing 250 mM mannitol, which is isotonic to the interior of the embryo (*Enyedi et al., 2013*). This treatment led to the expected abrogation of edema formation (*Figure 5a–d*). In addition and more surprisingly, it also completely rescued epidermal hyperplasia and aggregate formation (*Figure 5a–k*). Similar results were obtained by incubating mutant embryos in isotonic Ringer's solution (*Figure 5i*). This indicates that deregulation of the osmotic state in *psoriasis* mutants results in epidermal hyperplasia, and that the osmoregulatory function of *atp1b1a* is required for proper epidermal homeostasis.

But is the loss of this osmoregulatory function also sufficient to induce the epidermal defects? To test this, we incubated wild-type embryos in 3 mM ouabain, an inhibitor of the pumping function of the α-subunit. Treated embryos displayed numerous formerly reported phenotypes (*Figure 5—*

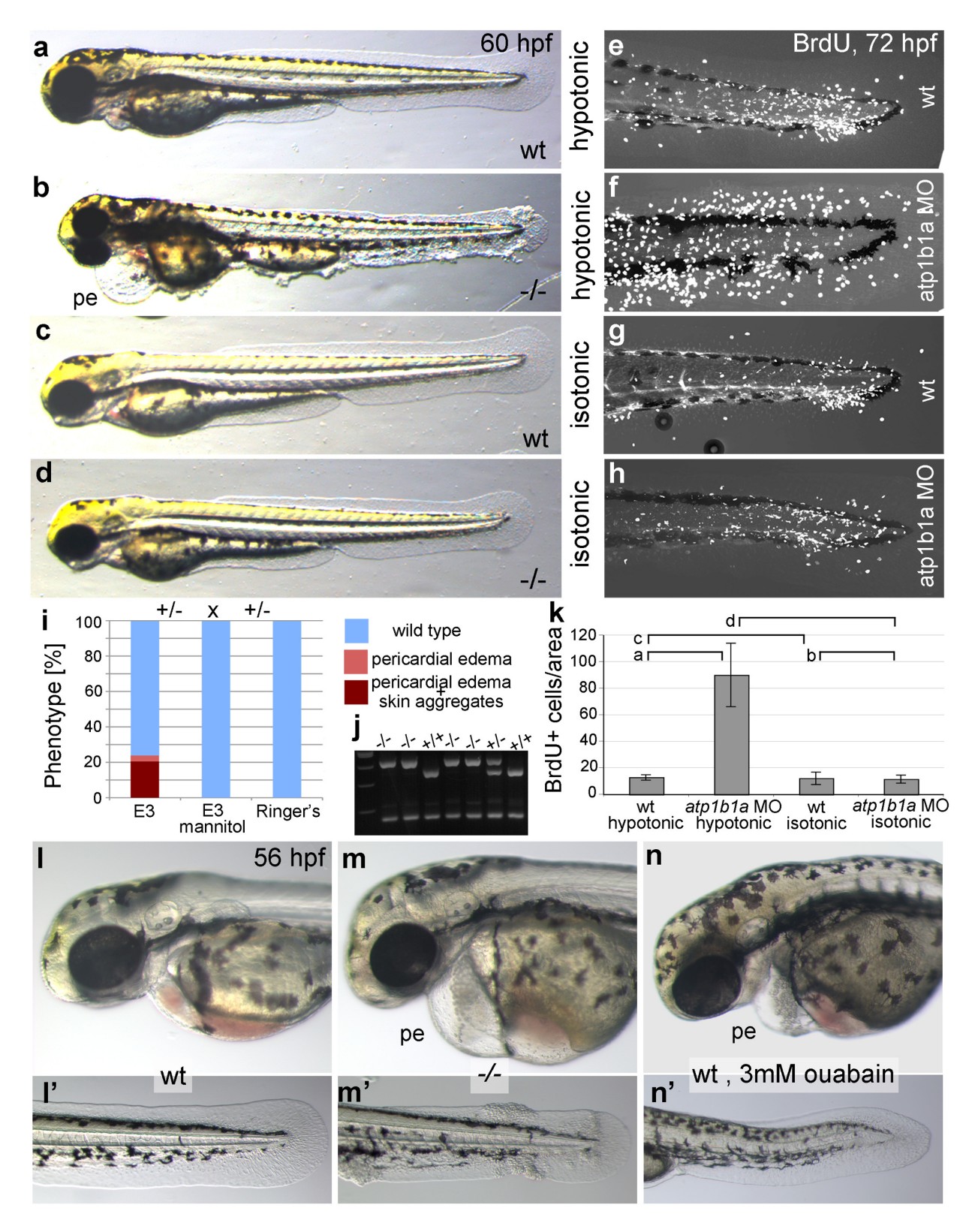

**Figure 5.** Epidermal hyperplasia is dependent on hypotonic conditions. (**a–k**) *psoriasis* mutants raised in isotonic conditions do not develop epidermal hyperplasia. Live images of 60 hpf embryos (**a–d**) show epidermal aggregates and pericardial edema (pe) in the mutant raised in hypotonic E3 (**b**) but

*Figure 5 continued on next page*

Figure 5 continued

not in the mutant raised in isotonic E3 (d). IF of BrdU incorporation (white) in 72 hpf wt or *atp1b1a* morphant tail fins (e–h) reveals excessive cell proliferation in the fin fold of the morphant embryo (f) raised in hypotonic E3 but not in the morphant embryo (h) raised in isotonic E3. (i) Quantification of embryos as shown in (a–d), obtained from incross of two *psoriasis* /- parents. n = 86–122. (j) Representative gel with PCR products subjected to *MwoI* restriction digest to genotype embryos raised in isotonic medium. All mutants raised in isotonic medium and shown as representative examples in this and the following figures had been positively genotyped. (k) Quantification of embryos as shown in (e–h), scored is the number of BrdU-positive cells in a given area of the median fin fold. n = 3–7 per condition. Error bars represent standard deviation. p values are as follows: a: 0.0416, b: 0.9139, c: 0.8956, d: 0.0017. (l–n) Live images of 56 hpf wt embryos (l,l'), *psoriasis* mutant embryos (m,m'), and wt embryos treated with 3mM ouabain (n,n') starting from 33 hpf. Blockage of Na,K-ATPase pump function by ouabain results in pericardial edema (pe) as in *psoriasis* mutants (n = 122/125; compare n to m), but not in epidermal aggregates (n = 0/125; compare n' to m').

The following source data and figure supplement are available for figure 5:

**Source data 1.** Source data for *Figure 5*.
**Figure supplement 1.** Live images of the otic vesicles of 48 hpf embryos show two otoliths in a untreated wt embryo (a) whereas in an embryo treated with 3 mM ouabain starting from 10 hpf, otoliths have failed to form (b; n = 76/99) or are much smaller (not shown; n = 23/99).

figure supplement 1; and data not shown), typical of those caused by genetic mutations in other Na,K-ATPases (*Blasiole et al., 2006*), as well as pericardial edema (*Figure 5l–n*). These embryos did not, however, develop epidermal aggregates (*Figure 5n'*, compare to *Figure 5m'*), indicating that the loss of the osmoregulatory function of Atp1b1a is necessary but not sufficient for epidermal hyperplasia. This points to an additional, pump-independent function of Atp1b1a, which makes keratinocytes more resistant to hypotonic stress.

## Atp1b1a is required for proper epithelial organization of epidermal cells independently of hypotonic conditions

Several functions additional to their role in ion homeostasis have been described for Na,K-ATPases, such as a role in epithelial polarity that promotes basolateral membrane identities, as well as a role in intercellular adhesion (see Introduction). To study the subcellular localization of zebrafish Atp1b1a, we transiently expressed an *atp1b1a-gfp* fusion construct in peridermal cells, one of its endogenous expression sites (*Figure 3i*), under the control of the *krt4* promoter. As described for Na,K-ATPases in other epithelial cells, Atp1b1a-GFP was localized in the basolateral domain of peridermal cells, together with E-cadherin (Cdh1), while it was absent from the apical region marked by the tight junction protein Tjp1/ZO-1 (*Kiener and Hunziker, 2007*) (*Figure 6a*). Consistent basolateral defects were found in *psoriasis* mutants even when incubated in isotonic medium to exclude effects resulting from tonicity-related stress. TEM and IF studies revealed unaltered morphologies of tight junctions (*Figure 6d,e*) and an unaltered distribution of Tjp1 in the periderm of mutant embryos at 52 hpf (*Figure 6—figure supplement 1*). By contrast, the lateral regions of peridermal cells were less organized (*Figure 6d,e*), and aberrant gaps were observed between peridermal cells and underlying basal keratinocytes (*Figure 6b–f*), Furthermore, the membranous localization of Cdh1 and Lgl2 proteins was strongly diminished in both the peridermal and basal cells of *psoriasis* mutants, independent of whether they had been incubated in hypotonic or isotonic medium (*Figure 7a–h*). At later stages (84 hpf), basal keratinocytes of *psoriasis* mutants displayed a strong and tonicity-independent reduction in normally basally localized cytokeratins (*Figure 7i–l*), similar to that previously reported for *lgl2* mutants (*Sonawane et al., 2005*). Together, this evidence points to a requirement for Atp1b1a for proper establishment of epithelial polarity and integrity, not only in the periderm but also in the basal layer.

## *atp1b1a* interacts with *lgl2* and is required in peridermal cells to establish proper epithelial organization of basal keratinocytes

In *lgl2* mutants, the epidermal defects only become apparent comparably late (from 96 hpf onwards) (*Reischauer et al., 2009*; *Sonawane et al., 2005*), but a recent study uncovered a pro-basal effect of *lgl2* on basal keratinocytes as early as 30–36 hpf (*Westcot et al., 2015*). This suggests that Atp1b1a might regulate epithelial polarity and epidermal integrity at least in part by cooperating with Lgl2 and by ensuring its proper localization. To test whether, in line with this notion, *lgl2* and

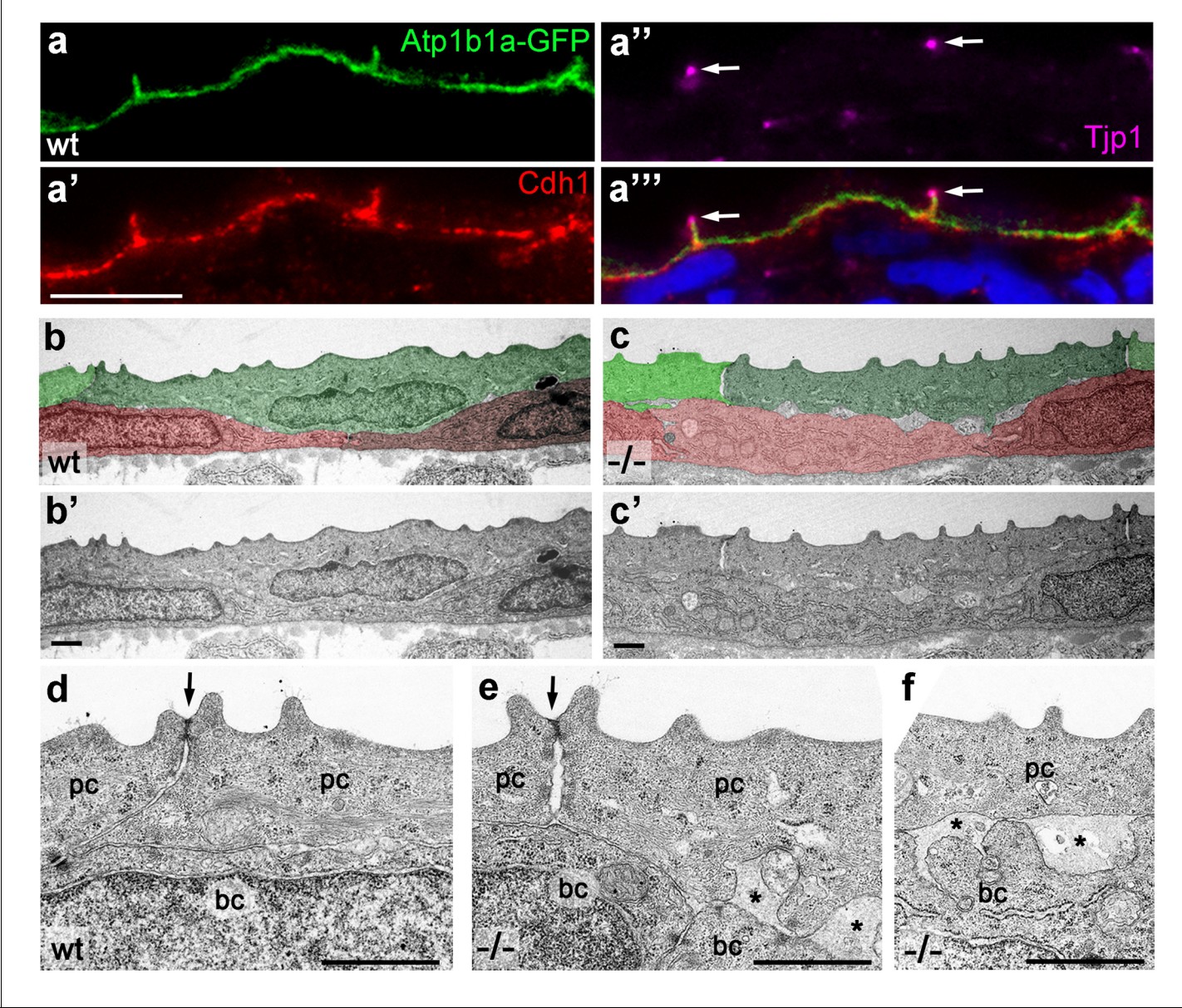

**Figure 6.** Atp1b1a is required for epidermal cell adhesion. (**a**) IF of GFP (**a**, green), Cdh1 (**a'**, red) and tight junction marker Tjp1 (**a''**, magenta) on a transverse section of the epidermis of a 48 hpf wt embryo expressing peridermal-specific *krt4:atp1b1a-gfp*, counterstained with DAPI (blue; **a'''** with merged channels). Atp1b1a and Cdh1 are co-localized on the basolateral side of peridermal cells, but are excluded from tight junctions and the apical side of these cells. (**b–f**). Transverse TEM sections through the medium fin fold of wt and *psoriasis* mutant embryos raised in isotonic conditions, at 58 hpf. In the mutant epidermis (**c,c'**), aberrant gaps between peridermal cells (false-colored in green) and underlying keratinocytes (false-colored in red) are apparent when compared to the wt epidermis (**b,b'**). (**d–f**) Higher magnifications reveal tight junctions (indicated by arrows) of unaltered morphology in the mutant (**e**) compared to a wt sibling (**d**), but less organized lateral regions between peridermal cells (**e**), and large gaps (*) between peridermal and basal cells (**e,f**) in the mutant. bc, basal cell; pc, peridermal cell. Scale bars: 1 μm.

The following figure supplement is available for figure 6:

**Figure supplement 1.** Localization of the tight junction protein Tjp1 is unaltered in *psoriasis* mutants.

*atp1b1a* genetically interact, we performed synergistic enhancement studies. When either *lgl2* or *atp1b1a* MOs were injected at a low, sub-phenotypic concentration, all embryos displayed wild-type morphology. By contrast, when the two MOs were co-injected at the same concentrations, 62% of

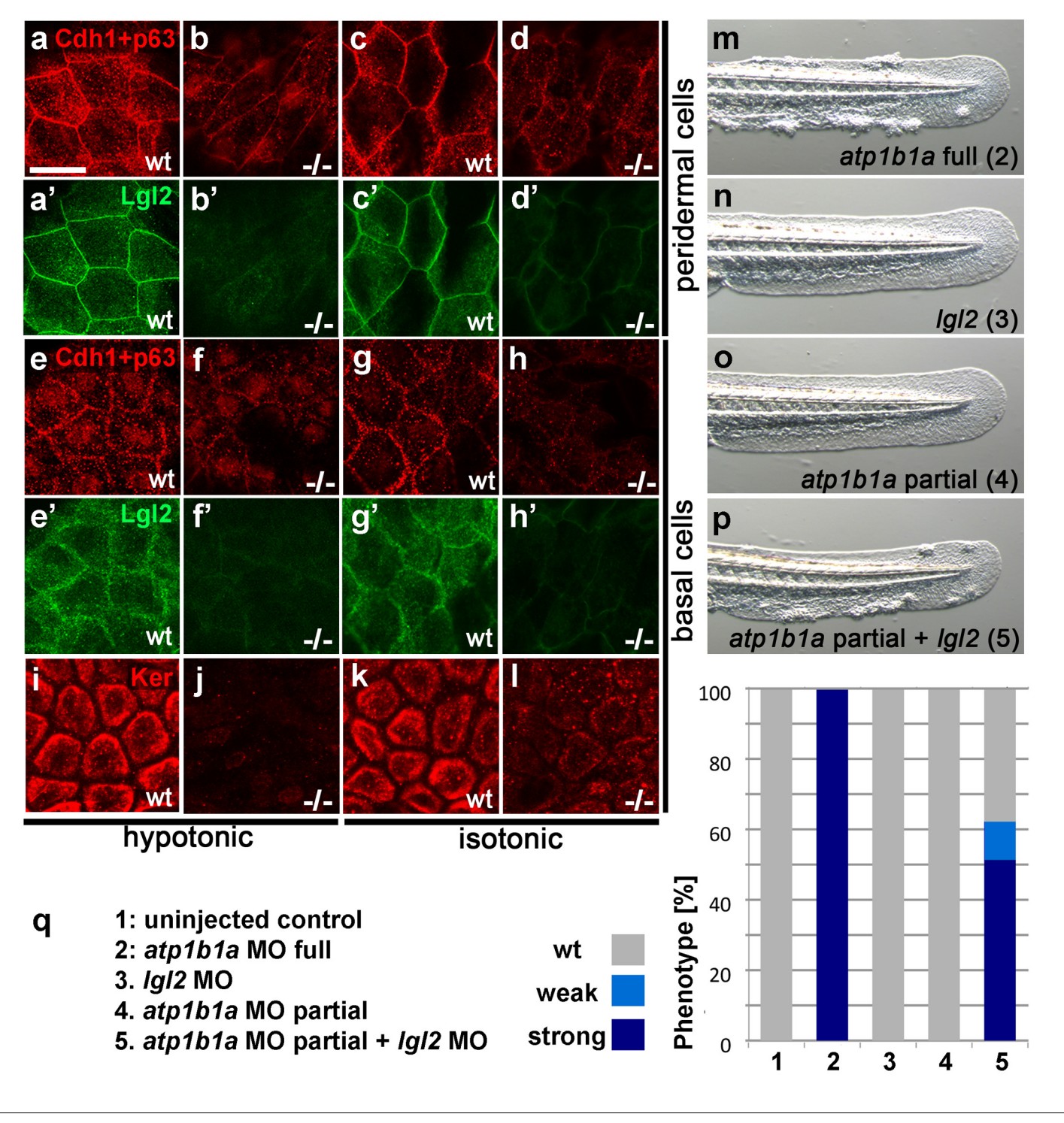

**Figure 7.** Atp1b1a is required for epidermal integrity and polarity. (a–h). Whole mount IFs of Cdh1 (red) and Lgl2 (green) in 54 hpf embryos raised in hypotonic (a,b,e,f) and isotonic (c,d,g,h) conditions. In mutants, localization of Cdh1 (b, d, f, and h; compare to wt a, c, e, g) and Lgl2 (b', d', f', and h'; compare to wt a', c', e', g') is compromised in both peridermal cells (a–d) and basal cells (e–h). Images show regions of the trunk epidermis not yet affected by aggregate formation. (i– l). Whole mount IFs of cytokeratin (red) in 84 hpf embryos show a reduction of basally localized cytokeratin in mutants raised in hypotonic (j) and isotonic (l) conditions. Images show regions of the trunk epidermis above the yolk sac extension. (m–p). Live images of the tail fins of 54 hpf embryos either with full MO knockdown of *atp1b1a* (m) or with partial MO knockdown of *lgl2* (n), of *atp1b1a* (o), or of both (p). (q) Quantification of the phenotypes of 54 hpf embryos in synergistic enhancement studies; n = 31–88. Similar results were obtained in two additional independent experiments.

*Figure 7 continued on next page*

*Figure 7 continued*

The following source data and figure supplements are available for figure 7:

**Source data 1.** Source data for *Figure 7q*.

**Figure supplement 1.** *atp1b1a* and *lgl2* interact genetically to enhance edema formation and AKT phosphorylation and mmp9 expression in basal keratinocytes of embryos raised in hypotonic medium .

**Figure supplement 1–source data 1.** Source data for *Figure 7—figure supplement 1j*.

the embryos (n = 74) developed pericardial edema, a loss of Na/K-ATPase α-subunits from the basolateral domains of pronephric epithelial cells, epidermal aggregates, and an upregulation of the malignancy markers pAKT and *mmp9* in basal keratinocytes, similar to the phenotype after full knockdown of *atp1b1a* (*Figure 7m–q*; *Figure 7—figure supplement 1*). This indicates that *lgl2* and *atp1b1a* cooperate to promote proper epithelial cell polarity and integrity in the epidermis as well as proper osmoregulation in the kidney, thereby suppressing epidermal malignancy.

In contrast to *lgl2* expression (*Sonawane et al., 2005*; *Sonawane et al., 2009*), *atp1b1a* expression in the epidermis is restricted to the periderm, suggesting that the effect of this gene on basal keratinocytes must be indirect. To prove this more directly, and to rule out indirect effects caused by hypotonicity, we analyzed chimeric embryos under isotonic conditions. Wild-type basal cell precursors that were transplanted into *atp1b1a* morphant hosts later exhibited distorted keratin distribution similar to that of their morphant neighbors (*Figure 8a*). Vice versa, *atp1b1a* morphant basal cells transplanted into wild-type hosts were indistinguishable from their wild-type neighbors and displayed normal keratin distribution (*Figure 8b*). This indicates that keratin localization in basal keratinocytes and the epithelial polarity of these cells are under the control of Atp1b1a function in a tissue other than the basal epidermal layer and the osmoregulatory organs. To show that this tissue is the periderm, we established a stable line of the aforementioned *krt4* transgene driving expression of *atp1b1a-gfp* exclusively in peridermal cells. In *psoriasis* mutants carrying this transgene that were raised under isotonic conditions, keratin localization in basal keratinocytes was fully restored and indistinguishable from that in wild-type siblings (*Figure 8c–e*; n = 28/28; 3 independent experiments). We conclude that *atp1b1a* is required in the periderm to establish proper epithelial polarity and organization in the underlying basal keratinocytes. Furthermore, this function is independent of the osmoregulatory role of Atp1b1a, as the corresponding defects are present even in mutants kept in isotonic medium, thus in the absence of hypotonic stress. When kept in hypotonic medium, *psoriasis* mutants carrying the *krt4:atp1b1a-gfp* transgene displayed a partial restoration of epidermal cell polarity both in the periderm and in the basal layer (cytokeratin and Lgl2 localization; *Figure 8—figure supplement 1e,l,m and f,j,n*) and a partial, but significant amelioration of basal cell malignancy (*Figure 8—figure supplement 1a–c*), including reduction in the expression of the malignancy markers pAKT and *mmp9* (*Figure 8f–h*; *Figure 8—figure supplement 1g,k,o*). The failure of full restoration under hypotonic conditions suggests that hypotonic stress further challenges the epidermal polarity system, which becomes more difficult to repair than is the case under isotonic conditions. In addition, the data reveal that upon hypotonic stress, the degrees of epidermal polarity defects and epidermal malignancy are proportionally linked.

## Malignant transformation of basal keratinoyctes is mediated via a PI3K-AKT-mTORC1-NFκB pathway

We next explored the molecular pathways mediating the malignant transformation within basal keratinocytes of *psoriasis* mutants. In cultured human keratinocytes, hypotonic culturing conditions induce, among other effects, the phosphorylation and activation of the kinase AKT/PKB (*Kippenberger et al., 2005*), while a PI3K-AKT-NFκB-MMP9 (metalloprotease 9) pathway has been implicated with cellular invasiveness (*Dilly et al., 2013*; *Wang et al., 2011*). Furthermore, AKT has been shown to activate NFκB via mTORC1 (a mechanistic target of rapamycin complex 1) (*Dan et al., 2008*).

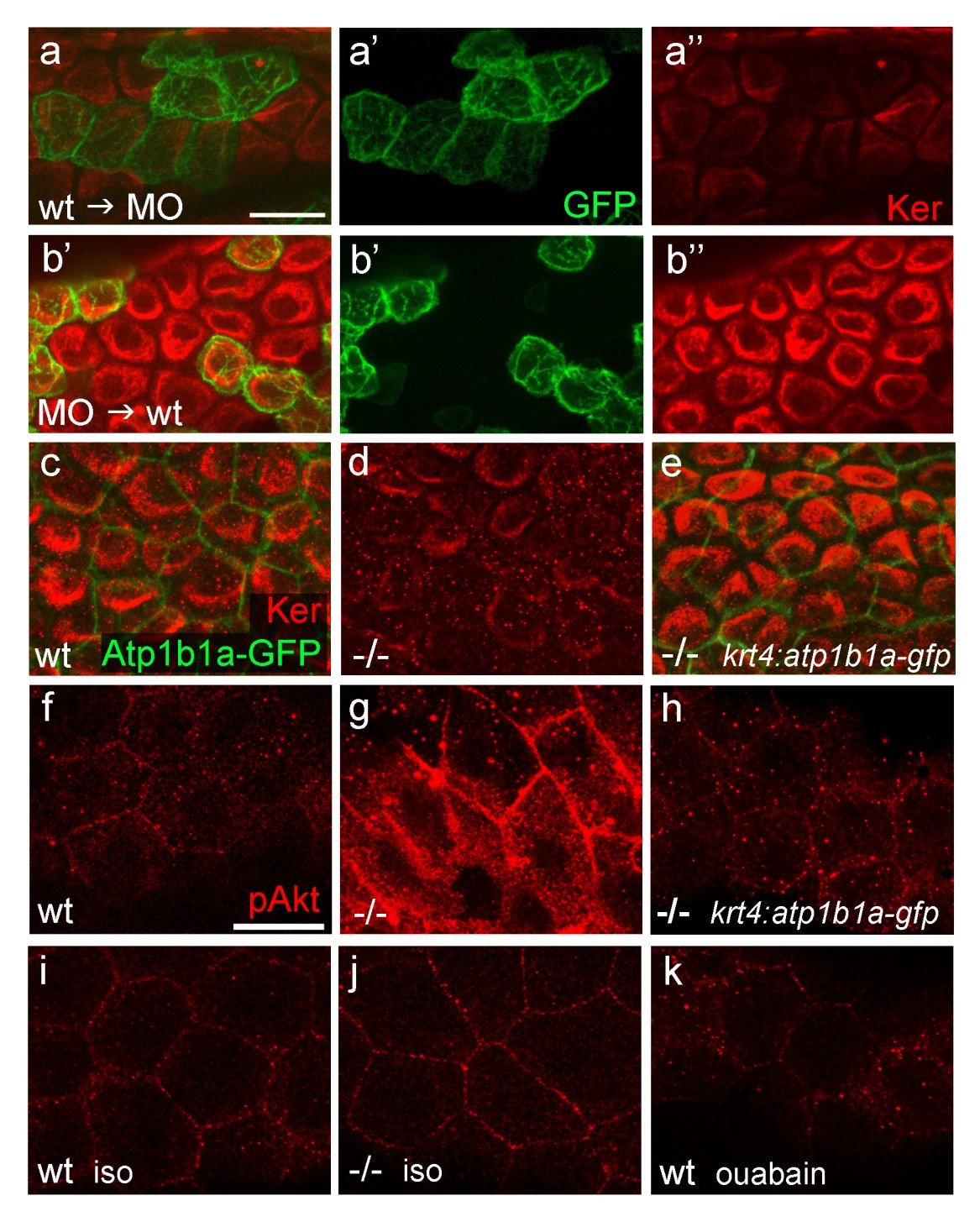

**Figure 8.** *atp1b1a* is required in peridermal cells to establish epithelial organization of basal keratinocytes, and to suppress hypotonicity-induced upregulation of pAKT levels in basal keratinocytes. (**a–b**) IF of cytokeratins (Ker; red) and GFP (green) in 84 hpf chimeric embryos raised in isotonic conditions. Wild-type (wt) basal keratinocytes expressing *Tg(Ola.Actb:Hsa.hras-egfp)-encoded* membrane-tagged GFP transplanted into *atp1b1a* morphant hosts display reduced cytokeratin localization similar to that of host cells (**a**), whereas the cytokeratin distribution of morphant donor cells (green) in wt hosts is indistinguishable from that in neighboring wt cells (**b**). Images show regions of the trunk epidermis above the yolk sac extension. (**c–e**). Maximum intensity projections of confocal images of IF of cytokeratin (red) and periderm-specific Atp1b1a-GFP (green) in 84 hpf embryos obtained from an in-cross of *psoriasis +/- ; Tg(krt4:atp1b1a-gfp)* parents, raised in isotonic conditions. Embryos were genotyped after imaging. Cytokeratin localization in basal keratinocytes is distorted in non-transgenic *psoriasis-/-* embryos (**d**; compare to wt in **c**), but restored in *psoriasis-/- ; Tg*

*Figure 8 continued on next page*

Figure 8 continued

*(krt4:atp1b1a-gfp)* embryos (e). Images show regions of the trunk epidermis above the yolk sac extension. (f–k) pAKT IF (red) in 54 hpf embryos. pAkt is upregulated in *psoriasis* mutants raised in hypotonic (g) but not in isotonic (j) medium, compared to wt siblings (f, i). pAkt levels are ameliorated in *psoriasis-/- ; Tg(krt4:atp1b1a-gfp)* embryos kept in hypotonic medium (h). pAkt is not upregulated in wt embryos incubated in hypotonic medium after addition of 3 mM ouabain, starting from 33 hpf (k). Images show regions of the trunk epidermis not yet affected by aggregate formation. Abbreviation: iso, isotonic medium (E3 250 mM mannitol).

The following source data and figure supplements are available for figure 8:

**Figure supplement 1.** *Tg(krt4:atp1b1a-gfp)*-driven *atp1b1a* expression in the periderm of *psoriasis* mutants kept in hypotonic medium leads to a partial rescue of the polarity defects and malignant transformation in basal keratinocytes.

**Figure supplement 1–source data 1.** Source data for *Figure 8—figure supplement 1a*.

*psoriasis* mutants that were raised in hypotonic conditions displayed strongly increased pAKT levels in both peridermal and basal cells, whereas wild-type siblings showed very little pAKT staining (*Figure 8f,g*). By contrast, pAKT levels were not elevated in *atp1b1a* mutants raised in isotonic medium (*Figure 8i,j*) or in wild-type embryos treated with ouabain (*Figure 8k*). Together, this evidence indicates that the pAKT pathway is only activated by hypotonic stress in conjunction with the epithelial polarity / adhesiveness defects caused by the loss of Atp1b1a function in the periderm.

To investigate the functional involvement of PI3K, mTORC1 and NFκB in malignant basal cell transformation of *psoriasis* mutants, we blocked them with the PI3K inhibitors Wortmannin, PIK90 or LY294002, the mTORC1 inhibitors Rapamycin or AZD8055, and the NFκB inhibitor Withaferin A. We determined the effects of these drugs on epidermal aggregate and edema formation, epidermal polarity (cytokeratin localization), pAKT levels, pS6RP levels (as a readout of mTORC1 activity) (*Hoesel and Schmid, 2013*), NFκB activity (transgenic NFκB responder line; *Candel et al., 2014*), expression levels of *mmp9* (which in mammals is a direct transcriptional NFκB target (*Rhee et al., 2007*) implicated in invasiveness), and finally epidermal proliferation. In addition to distorted keratin distribution (*Figure 9f,f'*) and increased pAKT levels (*Figure 9g,g'*) and proliferation rates (*Figure 9k,k'*) as described above (*Figures 5,7,8*), basal keratinocytes of *atp1b1a* mutants and morphants displayed strongly upregulated pS6RP levels (*Figure 9h,h'*), NFκB activity (*Figure 9i,i'*) and *mmp9* expression (*Figure 9j,j'*). Upon treatment with any of the inhibitors, mutants continued to display edema (*Figure 9a–e*) and distorted keratin distribution in basal keratinocytes (*Figure 9f''–f''''*), but they lacked epidermal aggregates (*Figure 9a–e*) and displayed *mmp9* expression levels (*Figure 9j''–j''''*) and keratinocyte proliferation rates (*Figure 9k''–k''''*) that were back to wild-type levels.

In addition, PI3K inhibition led to a downregulation and normalization of pAKT, pS6RP and NFκB activity levels (*Figure 9g'',h'',i''*), whereas the mTORC1 inhibitor affected the pS6RP and NFκB but not the pAKT levels (*Figure 9g''',h''',i'''*), and the NFκB inhibitor the NFκB but not the pS6RP and pAKT levels (*Figure 9g'''',h'''',i''''*). Together, this evidence points to the involvement of a linear PI3K-Akt-mTorC1-NFκB pathway in mediating epidermal malignancy downstream of the epidermal polarity and hypotonicity defects.

## MMP9 acts downstream of NFκB to mediate epidermal invasiveness, but not overgrowth

A normalization of epidermal hyperplasia to that described above was also obtained upon blockage of cell proliferation by incubating mutant embryos in 50 mM hydroxyurea; by contrast, edema (*Figure 10e*; *Figure 10—figure supplement 1a,b*), compromised cytokeratin localization and increased pAKT, NFκB activity and *mmp9* expression levels persisted (*Figure 10a–d*; *Figure 10—figure supplement 1c*; and data not shown). By contrast, *mmp9* knockdown via MO injection failed to rescue epidermal hyperplasia (*Figure 10e*), but alleviated epidermal invasiveness. Thus, the degradation of the skin basement membrane was significantly, but not fully, blocked in *mmp9*-MO-injected *psoriasis* mutant embryos (*Figure 10f,g*, *Figure 10—figure supplement 2*). In addition, basal keratinocytes remained within the epidermal compartment (*Figure 10h*), in contrast to their

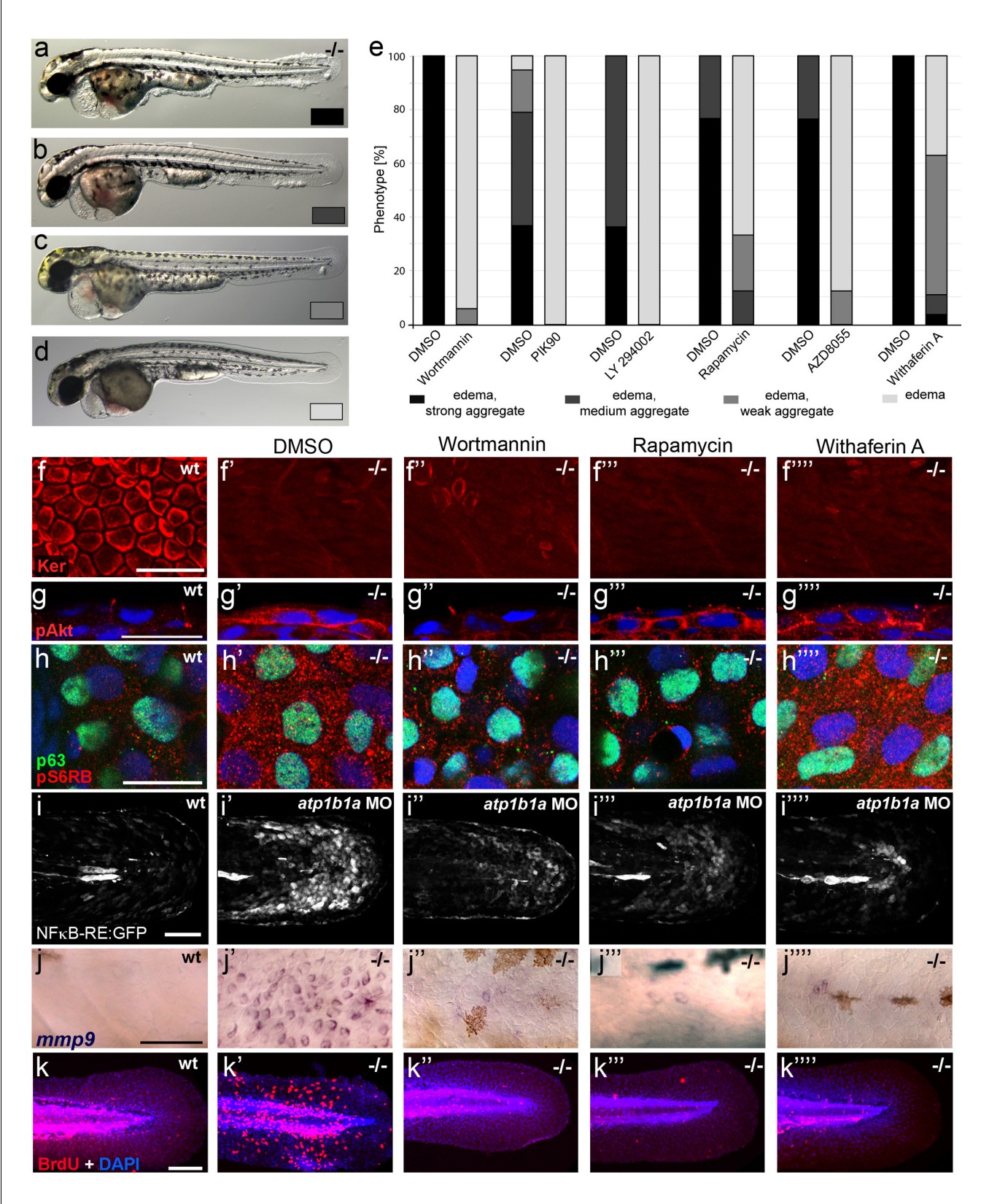

**Figure 9.** Hyperplasia and transcriptional upregulation of *mmp9* in basal keratinocytes of *psoriasis* mutants is mediated via an aberrant activation of a linear PI3K-Akt-mTorC1-NFκB pathway. (a–e) Blockade of PI3K, mTorC1, and NFκB signaling rescues epidermal aggregate but not pericardial edema
*Figure 9 continued on next page*

*Figure 9 continued*

formation in *psoriasis* mutants. (**a–d**) Representative live images of phenotypic strength classes of *psoriasis -/-* embryos at 54 hpf, all with pericardial edema of comparable strengths, but strong (**a**), intermediate (**b**), weak (**c**), or no (**d**) epidermal aggregates. (**e**) Quantification of the phenotypes of *psoriasis* mutants incubated in E3 medium containing 1 µM Wortmannin, 5 µM PIK90, 25 µM LY94002, 1.1 µM Rapamycin, 30 µM AZD8055, or 30 µM Withaferin A compared to the corresponding DMSO controls (n = 16–30). Drugs were added at 34 hpf and embryos scored at 54 hpf. Similar results were obtained in at least two additional independent experiments. For representative live images, see *Figure 9—figure supplement 1*. f–k. A linear PI3K-Akt-mTORC1-NFκB pathway mediates hyperplasia and upregulation of *mmp9* expression in basal keratinocytes. All embryos had been kept in (hypotonic) E3 medium, supplemented with the indicated drugs starting at 34 hpf. (**f–f''''**) IF of cytokeratins (red) at 84 hpf. Distorted keratin localization in the *psoriasis* mutant (**f'**, compare to wt (**f**)) is not restored by Wortmannin (**f''**), Rapamycin (**f'''**), or Withaferin A (**f''''**). Scale bar: 50 µm. (**g–g''''**) IF of pAkt (red), counterstained with DAPI (blue); transverse sections of 54 hpf *psoriasis* mutants raised in E3 medium. Elevated pAkt levels in the mutant (**g'**, compared to the wt (**g**)) are lowered by Wortmannin (**g''**), but not by Rapamycin (**g'''**) or Withaferin A (**g''''**). Scale bar: 20 µm. (**h–h''''**) IF of pS6RP and p63 of whole mounts, at 54 hpf. Elevated pS6RP levels in mutant (**h'**, compared to wt (**h**)) are alleviated by Wortmannin (**h''**), PIK90 (not shown) and Rapamycin (**h'''**), but not by Withaferin A (**h''''**). Scale bar: 20 µm. (**i– i''''**) Confocal images of GFP fluorescence in the tail fin of a live 48 hpf wt embryo (**i**) and an *atp1b1a* morphant (**i'**), both carrying the *Tg(NFκB-RE:eGFP)* transgene. The *atp1b1a* morphant shows strong upregulation of NFκB activity in keratinocytes, which is restored by treatment with Wortmannin (**i''**), Rapamycin (**i'''**), and Withaferin A (**i''''**). Scale bar: 100 µm. For quantification, see *Figure 9—figure supplement 2*. (**j–j''''**) *mmp9* WISH at 54 hpf. Elevated *mmp9* expression in the mutant epidermis (**j'**, compare to wt (**j**)) is downregulated by Wortmannin (**j''**), Rapamycin (**j'''**), and Withaferin A (**j''''**). Scale bar: 50 µm. (**k–k''''**) IF of incorporated BrdU (red), counterstained with DAPI (blue) at 56 hpf. Elevated cell proliferation in mutant epidermis (**k'**, compare to wt (**k**)) is downregulated by Wortmannin (**k''**), Rapamycin (**k'''**), and Withaferin A (**k''''**). Scale bar: 100 µm.

The following source data and figure supplements are available for figure 9:

**Source data 1.** Source data for *Figure 9e*.

**Figure supplement 1.** Chemical inhibiton of PI3K, mTORC1 or NFkB rescues the epidermal malignancies, but not the pericardial edema of *psoriasis* mutants.

**Figure supplement 2.** Quantification of NFκB activity in wt embryos, *atp1b1a* morphants and *atp1b1a* morphants treated with Wortmannin, Rapamycin, and Withaferin A, as shown in *Figure 9i–i''''*.

**Figure supplement 2–source data 1.** Source data for *Figure 9—figure supplement 2*.

ectopic localization in dermal compartments of mutant controls (*Figure 1j*). This indicates that Mmp9 is an essential downstream effector of the PI3K-AKT-NFκB pathway and is primarily involved in mediating epidermal invasiveness, whereas epidermal hyperplasia is mediated by other NFκB targets.

## Discussion

### The role of Atp1b1a in kidney and heart to avoid systemic hypotonicity

*atp1b1a* is expressed in multiple organs and epithelia, including the periderm of the epidermis, the tubules / ducts of the pronephros and the heart (*Figure 3*). Accordingly, *atp1b1a* mutant embryos display compromised heart function (*Figure 4*), consistent with results reported in mouse (*Barwe et al., 2009*). In addition, they display a loss of the α-subunit of the Na/K-ATPase, the actual pump, from the basolateral domains of pronephric epithelial cells (*Figure 4*), possibly reflecting more general defects in the cells' epithelial polarity and severely compromising the kidney's osmo-regulatory function. Together with the reduction in the blood pressure, these defects should lead to compromised water elimination in the pronephros, as reflected by the reduced clearance of rhoda-min-dextran from the blood (*Figure 4*). The resulting increase in water content in turn causes edema formation and a corresponding reduction in the osmolarity of interstitial compartments (*Drummond et al., 1998*; *Hentschel et al., 2005*).

### The role of Atp1b1a during epithelial polarity and cell-cell adhesion in the epidermis, and its trans-effect from the periderm to the basal layer

In addition, and independently of its osmoregulatory functions, Atp1b1a is required in the periderm to assure proper epithelial polarity and cell-cell adhesiveness, both in the periderm and in the

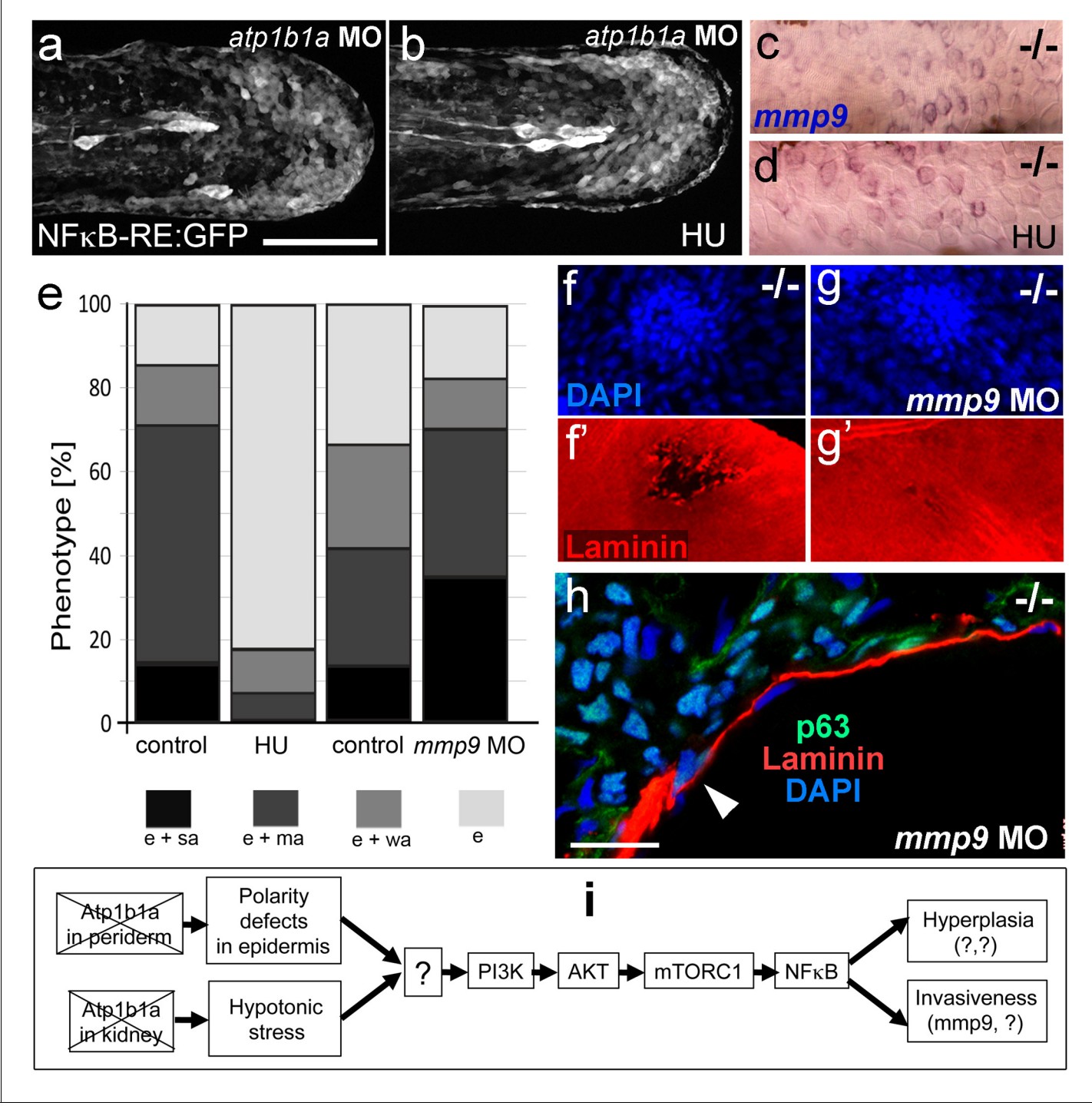

**Figure 10.** Blockage of cell proliferation results in the normalization of epidermal hyperplasia, whereas blockage of Mmp9 activity reduces epidermal invasiveness in *psoriasis* mutants. (a–b).Confocal images of GFP in the tail fin of live 48 hpf *atp1b1a* morphant *Tg(NFκB-RE:eGFP)* transgenics, showing that elevated NFκB activity in *atp1b1a* morphants (a) is not reduced by hydroxyurea treatment (b). For quantification, see *Figure 10—figure supplement 1*. (c–d) *mmp9* WISH of 54 hpf *psoriasis* mutants raised in hypotonic E3. Elevated *mmp9* expression in mutant epidermis (c) is not reduced by hydroxyurea (HU) treatment (d). (e) Quantification of the phenotypes of *psoriasis* mutants, either treated with 50 mM hydroxyurea or injected with *mmp9* MO, compared to their respective siblings. e, pericardial edema; wa, weak epidermal aggregates; ma, medium epidermal aggregates; sa, strong epidermal aggregates . n = 17–47. Similar results for each condition were obtained in two additional independent experiments. (f–g) *mmp9* knockdown alleviates basement membrane fragmentation. Laminin IF, counterstained with DAPI, in *psoriasis* mutants at 58 hpf, epidermal aggregates of comparable sizes. In the un-injected *psoriaris* mutant (f), the aggregate is associated with BM fragmentation, while the underlying BM is largely intact

*Figure 10 continued on next page*

*Figure 10 continued*

in the *psoriasis* mutant injected with *mmp9* MO (**g**). For more images and numbers, see *Figure 10—figure supplement 2*. (**h**) *mmp9* knockdown alleviates epidermal invasiveness. Laminin and p63 IF of transverse sections, counterstained with DAPI, through the yolk sac of a *psoriasis* mutant (58 hpf) injected with *mmp9* MO. The basement membrane is largely intact (arrowhead to small remaining region with thinner basement membrane), and p63 keratinocytes are confined to the epidermal compartment above the basement membrane. For un-injected mutant and wt controls, see *Figure 2i,j*. (**i**) Diagram of the identified pathway in which the two required non-cell-autonomous effects caused by loss of Atp1b1a in periderm and osmoregulatory organs converge in basal cells. The pathway subsequently diverges downstream of NFκB to mediate overgrowth versus invasiveness of transformed keratinocytes. Question marks indicate components that have not yet been identified. For details, see text.

The following source data and figure supplements are available for figure 10:

**Source data 1.** Source data for *Figure 10e*.

**Figure supplement 1.** Morphology rescue of *psoriasis* mutant upon treatment with hydroxyurea and quantification of the non-alleviating effect of the treatment on NFκB activity in embryos as shown in *Figure 10a,b*.

**Figure supplement 1—source data 1.** Source data for *Figure 10—figure supplement 1*.

**Figure supplement 2.** *mmp9* knockdown alleviates epidermal invasiveness.

---

underlying basal keratinocytes. Thus, even when the osmoregulatory role of Atp1b1a has become dispensable upon incubation in isotonic medium, peridermal cells of mutant embryos display reduced levels of E-cadherin and of Lgl2, a pro-basal polarity regulator, as well as increased spacing between epidermal cells. Strikingly, basal keratinocytes themselves also display specific defects, such as reduced membraneous Lgl2 and E-cadherin levels, as well as a mislocalization of cytokeratins in basal domains. These defects could be rescued when transgene-encoded wild-type Atp1b1a was re-introduced into peridermal cells, indicating that Atp1b1a from the periderm regulates the epithelial organization and adhesiveness of basal keratinocytes in a tonicity-independent manner (*Figure 8*).

But what are the molecular mechanisms that underlie this trans-layer effect? Trans-bonds between ATPase β-subunits, previously reported to mediate cell-cell adhesiveness in other instances (*Vagin et al., 2012*), could account for the adhesiveness among peridermal cells, but not for that between peridermal and basal cells, as the latter lack *atp1b1a* expression. The cell-cell adhesion molecule E-cadherin might be the relevant mediator. In *psoriasis* mutants, E-cadherin membrane localization is severely diminished both in peridermal and in basal cells (*Figure 7*). The reduction of E-cadherin in peridermal cells could be a direct consequence of the loss of Atp1b1a, with which it is normally co-localized in basolateral regions (*Figure 6*). This is consistent with the formerly reported physical interaction between ATPase α-subunits and E-cadherin via the anchoring protein ankyrin and the spectrin-actin cytoskeleton (*Vagin et al., 2012*). In addition, Atp1b1a could promote E-cadherin localization via other epithelial polarity regulators, consistent with reports in MCDK cells (*Qin et al., 2005*). We propose that it is this loss of proper E-cadherin in peridermal cells that compromises not only their adhesion to the underlying basal cells (due to the loss of trans-bonds between the basal side of peridermal and the apical side of basal cells; *Figure 1*) but also the adhesion among basal cells themselves, as well as their epithelial polarity. Consistent with the latter notion, loss of E-cadherin has been shown to affect apical-basal polarity in numerous systems (*Desai et al., 2009*; *Stephenson et al., 2010*).

## Atp1b1a as a multi-functional non-cell autonomous tumor suppressor

E-cadherin and epithelial polarity regulators, including Lgl2, are well-known tumor suppressors (*Birchmeier, 1995*; *Martin-Belmonte and Perez-Moreno, 2012*; *Ellenbroek et al., 2012*). Atp1b1a is another cell adhesion molecule and epithelial polarity regulator to add to the list,, as its loss causes crucial features of carcinogenesis (hyperplasia and infiltration of other tissues) in vivo. Strikingly, however, epidermal malignancy only occurs upon the combined loss of Atp1b1a's osmoregulatory and epithelial polarity-regulating functions. Thus, hypotonic stress per se does not lead to basal cell malignancy, as demonstrated in this work by the treatment of wild-type embryos with ouabain. Consistently, none of the described kidney mutants, although displaying massive edema

formation as a result of internal hypotonicity, have had reported skin defects (*Drummond et al., 1998*; *Drummond, 2002*; *Hentschel et al., 2005*). Moreover, epidermal malignancy in *atp1b1a* mutants is suppressed in the absence of hypotonic stress, although epithelial polarity and adhesiveness in the embryonic skin remain compromised.

Carcinogenesis is commonly regarded as a multistep process, to which aberrant polarity signaling can contribute as one of multiple causative factors (*Sherr, 2004*; *Sun and Yang, 2010*; *Ellenbroek et al., 2012*). In this light, carcinogenesis subsequent to the combined loss of different functions of one and the same factor is an astonishing and thus far unrecognized variation of this common concept. In addition, it is remarkable that one of these genetically caused effects (hypotonicity) can also be achieved by environmental insults (injury), constituting an interesting variation of the common concept of tumorigenesis as a result of combined genetic and environmental factors.

## The cooperation between Atp1b1a and Lgl2

Another epithelial polarity regulator that has a tumor-suppressing function in the epidermis of zebrafish embryos is Lgl2. Like *atp1b1a* mutants, *lgl2* mutants display aberrant apical-basal polarity of basal keratinocytes (mislocalized cytokeratin) and E-cadherin displacement, as well as epidermal hyperplasia and increased expression of *mmp9* (*Sonawane et al., 2005*; *Reischauer et al., 2009*), which we now can interpret as a sign of invasiveness. Although not yet addressed, carcinogenesis in *lgl2* mutants might also involve hypotonic stress, and *lgl2*, like *atp1b1a*, might act both in the epidermis and in the kidney. Thus, we found that *atp1b1a* and *lgl2* not only genetically interact during epidermal cell polarity and malignancy but also during edema formation and Na/K-ATPase α-subunit localization (*Figure 7—figure supplement 1*), in line with previous reports of pronephric defects and mild edema formation in *lgl2* morphants (*Tay et al., 2013*). However, *atp1b1a* and *lgl2* mutants also display differences. Although in conjunction with other mutations, loss of Lgl2 function leads to defects in earlier developmental stages (*Westcot et al., 2015*; this work), epidermal hyperplasia in *lgl2* single mutants begins significantly later (4 days post fertilization; dpf) than that in *atp1b1a* mutants (2 dpf). An even more pronounced requirement in different time windows has been reported for Na,K-ATPase and Lgl during epithelial polarity regulation in *Drosophila* embryos. Thus, defects in *lgl* mutants are manifested during gastrulation, when ATPase is dispensable, whereas corresponding defects in ATPase mutants are first detected during early organogenesis stages, when the defects in *lgl* mutants begin to recover (*Laprise et al., 2009*; *Laprise and Tepass, 2011*). This suggests that as in flies, Atp1b1a and Lgl2 in zebrafish might act in different basolateral-promoting complexes, which have differential temporal but partially redundant functions. In addition, although eventually converging at similar or even identical (*mmp9*) effector genes, Atp1b1a and Lgl2 can fulfill their tumor-suppressing role by blocking different pathways. Lgl2 has been reported to act by blocking ErbB and MAPK in later epidermal malignancies (*Reischauer et al., 2009*), but earlier it cooperates with Atp1b1a to inhibit aberrant activation of a PI3K-AKT-mTORC1-NFκB pathway (this work).

## The PI3K-AKT-mTORC1-NFκB pathway as an integrator of hypotonic stress and aberrant epithelial cell polarity

Of note, chemical inhibition of the PI3K-AKT-mTORC1-NFκB pathway in *atp1b1a* mutants blocks epidermal malignancy, but it does not rescue the edema or epithelial polarity phenotype. This indicates that the pathway acts downstream of, and integrates, these two primary effects caused by the loss of Atp1b1a, which per se are necessary but not sufficient for pathway activation (*Figure 10m*).

Parts of the PI3K-AKT-mTORC1-NFκB pathway had previously been reported to mediate hypotonic stress or tumorigenesis in other contexts. Thus, hypotonic stress induces the activation of the EGF receptor (EGFR), AKT and several MAP kinases (ERK1/2, p38) in cultured human keratinocytes (*Kippenberger et al., 2005*), in line with our observed increase in pAKT levels (but not in pERK levels; JH and MH, unpublished data). Furthermore, PI3K, AKT and NFκB mediate the tumorigenic effects of different cytokines in cultured prostate, gastric and leukemic cancer cells (*Dilly et al., 2013*; *Kang et al., 2011*; *Wang et al., 2011*), while mTORC1 controls NFκB activity downstream of pAKT in prostate and breast cancer cells (*Dan et al., 2008*; *Davis et al., 2014*).

The effects of the transcription factor NFκB in the context of tumorigenesis are complex, and multiple mechanisms and effectors have been described. NFκB can affect cancer cell survival, proliferation and invasiveness, and can act in cancer cells themselves or on the tumor microenvironment, for instance by regulating tissue inflammation, which in turn further stimulates tumor progression (*Ben-Neriah and Karin, 2011*; *Hoesel and Schmid, 2013*). In *atp1b1a* mutants, however, inflammation seems to be of minor relevance: epidermal aggregates only display moderately increased numbers of innate immune cells, and genetic ablation of the entire myeloid lineage does not alleviate epidermal malignancy (*Figure 2—figure supplement 2*). One prominent direct transcriptional target of NFκB is *mmp9* (*Rhee et al., 2007*), a matrix-metalloprotease with collagenase activity that destabilizes basement membranes and connective tissue, thereby facilitating tumor progression and metastasis (*Kessenbrock and Werb, 2010*; *Kang et al., 2011*; *Dilly et al., 2013*). In addition, MMPs can promote tumor vascularization and inflammation as well as tumor cell proliferation (*Kessenbrock and Werb, 2010*). In zebrafish *atp1b1a* mutants, however, the NFκB-dependent increase in *mmp9* expression only contributes to epidermal invasiveness, while epidermal hyperplasia is mediated by NFκB targets other than *mmp9* (*Figure 10m*). Thus, in contrast to *mmp9* knockdown, treatment of mutant embryos with hydroxyurea rescues epidermal hyperproliferation in the persistent presence of high NFκB activity. The nature of these proliferation-promoting NFκB targets remains unknown. CyclinD1 (*ccnd1*), a described direct transcriptional NFκB target in mammals (*Hinz et al., 1999*), seems an unlikely target, as its expression levels are unaltered in *atp1b1a* mutants (JH and MH, unpublished data).

What also remains unknown is the player upstream of PI3K that is directly affected by hypotonicity and by the compromised epithelial polarity / adhesiveness of basal keratinocytes (*Figure 10m*). EGFR would be a strong candidate. It acts as the upstream regulator of PI3K to mediate hypotonic stress in human keratinocytes (*Kippenberger et al., 2005*), and is activated by hypotonicity in several other, MAPK-mediated responses (*Lezama et al., 2005*). Like Atp1b1a, EGFR is normally targeted to the basolateral side of epithelial cells (*He et al., 2002*), and its signaling strength is under the control of the epithelial polarity system (*Hobert et al., 1999*; *Vermeer et al., 2003*). Nevertheless, treatment of *atp1b1a* mutant embryos with the chemical pan-ErbB inhibitor PD168393 failed to rescue epidermal malignancies (JH and MH, unpublished data). This suggests that other receptor tyrosine kinases or their modifiers are the relevant players at the convergence point of hypotonic stress and aberrant epithelial polarity signaling.

## Na,K-ATPases and hypotonicity during human carcinogenesis

Na,K-ATPases and hypotonicity might also play a role during human carcinogenesis. Although not usually mentioned in this context (*Martin-Belmonte and Perez-Moreno, 2012*; *Ellenbroek et al., 2012*), one study has proposed the β1-subunit as a potential tumor-suppressor (*Inge et al., 2008*). This notion was based on functional analyses performed with virus-transformed MDCK cells, and the down-regulation of the β1-subunit in different human carcinoma cell lines (*Espineda et al., 2004*), and is strongly supported by our genetic in vivo data presented here.

There are also several indirect lines of evidence for a tumor-promoting effect of hypotonic stress during human carcinogenesis. For instance, untransformed human keratinocytes display strongly increased proliferation when cultured in hypotonic medium (*Gönczi et al., 2007*). Systemic hypotonicity can also occur in humans in vivo. A major human condition that leads to systemic hypotonicity in conjunction with reduced Na levels (hypotonic or hypoosmolar hyponatremia) is increased nephric water re-absorption caused by ADH (antidiuretic hormone) hyperactivity. In SIADH (Syndrome of inappropriate antidiuretic hormone secretion), this is usually linked to carcinogenesis and results from ectopic ADH production by the tumors, which in general are rather aggressive (*Ellison and Berl, 2007*; *Grohé and Berardi, 2015*). In fact, hyponatremia is quite common in malignant solid tumors beyond SIADH (up to 25% of all patients). It is used as a prognostic and predictive value and is associated with high morbidity rates; its early diagnosis and treatment significantly improves the patients' prognosis (*Schutz et al., 2014*; *Grohé and Berardi, 2015*; *Balachandran et al., 2015*). This correlation suggests that systemic hypotonicity / hyponatremia increases carcinoma incidence and aggressiveness. In addition, the carcinogenesis risk seems to be increased by locally restricted hypotonic stress. Such stress might occur, for instance, during injuries of the lung or the esophagus (*Ribeiro et al., 1996*; *Islami et al., 2009*; *Goldkorn and Filosto, 2010*; *Maret-Ouda et al., 2016*), when epithelial cells become exposed to the airway surface fluid

(*Joris and Quinton, 1993*) or to saliva (*Edgar, 1992*), respectively, both of which are hypotonic compared to the internal milieu. In this light, and in light of the data presented in this work, it makes sense to include treatments that target osmotic conditions in therapies against certain carcinoma types (*Balachandran et al., 2015*; *Grohé and Berardi, 2015*). Furthermore, lavages with distilled water during cancer surgery (*Iitaka et al., 2012*), or hypotonic approaches to improve the uptake of chemotherapeutics by tumor cells (*Stephen et al., 1990*), should be used with caution.

## Materials and methods

### Zebrafish lines and genotyping

The mutant line *psoriasis*[m14] (*Webb et al., 2008*) and the transgenic lines *Tg (Ola.Actb:Hsa.hras-egfp)*[vu119Tg] (ubiquitous expression of membrane-tagged EGFP) (*Cooper et al., 2005*),*Tg(krt4: GFP)*[gz7Tg] (periderm-specific expression of GFP) (*Gong et al., 2002*), and *Tg(NFκB-RE:eGFP)*[sh235Tg] (NFκB responder) (*Candel et al., 2014*) have been described previously.

The *krt4:atp1b1a-gfp* construct was generated using the Tol2 kit (*Kwan et al., 2007*) with the described *krt4* promoter (*Gong et al., 2002*). The primers 5'- GGGGACAAGTTTGTACAAAAAAG-CAGGCTCCACCATGCCCGCAAATAAAGATGG-3' and 5'-GGGGACCACTTTGTACAAGAAAGC TGGGTATGACTTGGTTTTGATGGTGAAC-3' were used to amplify the *atp1b1a* cDNA, which was cloned into pDONR221 (Invitrogen, Carlsbad, CA). The construct was used to generate the stable transgenic line *Tg(krt4:atp1b1a-gfp)*[fr36Tg] by standard injection and screening procedures.

Genotyping of the *psoriasis* mutation was conducted by PCR with the primers 5'-TCCGAGAA TCCAAAATGAGC-3' and 5'-CACTCGTCTCCGTTTATTCG-3' followed by an *Mwo*I digestion of the PCR product.

Embryos were raised in E3 medium (5 mM NaCl, 0.17 mM KCl, 0.33 mM CaCl$_2$, 0.33 mM MgSO$_4$; hypotonic), E3 medium containing 250 mM mannitol, or Ringer's solution.

All zebrafish experiments were approved by the national animal care committees (LANUV Nordrhein-Westfalen; 8.87–50.10.31.08.129; 84–02.04.2012.A251; City of Cologne; 576.1.36.6.3.01.10 Be) and the University of Cologne.

### Whole exome sequencing and mapping

Genomic DNA was extracted from a pool of 20 affected embryos and from their healthy parents using a Maxwell 16 instrument from Promega, according to the manufacturer's protocol. The DNAs underwent individual library preparation and enrichment (SureSelectXT Zebrafish Kit 5190–5450, Agilent Technologies, Santa Clara, CA), using 3ug DNA fragmented to 150bp by sonication (bioruptor, Diagenode, Liège, Belgium) and the standard protocol SureSelectXT Target Enrichment for Illumina Paired-End Multiplexed Sequencing. After validation (Agilent 2200 TapeStation) and quantification (Invitrogen Qubit System), we performed a qPCR by using the Peqlab KAPA Library Quantification Kit and the Applied Biosystems 7900HT Sequence Detection System (Applied Biosystems, Foster City, CA). Pools of 2–4 libraries were sequenced on one lane using an Illumina TruSeq PE Cluster Kit v3 and an Illumina TruSeq SBS Kit v3-HS on an Illumina HiSeq2000 sequencer with a paired-end (101x7x101 cycles) protocol. The alignment to the zebrafish reference version Zv9 was performed using the Burrows-Wheeler Aligner (BWA). PCR duplicate marking was performed using Picard, both realignment around indels and variant calling were performed using the Genome Analysis ToolKit (GATK). The annotation was performed using Annovar and the variations were filtered according to the predicted effect at the protein level and according to their presence in a control set of 5 unrelated zebrafish Whole-Exome Sequencing datasets. For mapping, all variant loci with a coverage of at least 20 were selected from both the affected pool and the parental DNA, with the exclusion of loci in which both datasets were homozygous for the variant allele. At every locus, the percentage of reads showing the variation in the parental DNA was subtracted from the percentage of reads showing the variation in the affected pool. The absolute value of this difference was then plotted against the physical position of the locus and heatmaps were generated for the whole genome and for each chromosome. The chromosomes were then visually analyzed to identify the regions in which the majority of the loci show a difference of either 25% or 50%.

## Whole-mount in situ hybridization (WISH)

Embryos were fixed in 4% paraformaldehyde (PFA) and WISH was performed as previously described (*Carney et al., 2007*). DIG-labeled probes were synthesized with the Roche digoxygenin RNA synthesis kit, using *cmlc2*, *mmp9* and *mpx* cDNA templates as described (*Carney et al., 2007*; *Reischauer et al., 2009*). For *atp1b1a*, a cDNA was amplified using the following primers 5'-A TGCCCGCAAATAAAGATGG-3' and 5'-TCATGACTTGGTTTTGATGG-3', cloned into pGEM T Easy (Promega) and linearized with *Sac*II for Sp6 RNA pol-dependent antisense RNA synthesis. Combined colorimetric WISH and immunostainings were performed as described (*Carney et al., 2007*). Images were taken on a Axioplan2 microscope (Zeiss) using AxioVision software (Zeiss).

## Rhodamin-dextran injections

1.5 nl of 1 mg/ml rhodamine-dextran (Molecular Probes, Eugene, OR; D1816) in PBS was injected into the common cardinal vein of 34 hpf embryos anaesthetized with Tricaine. Injected embryos were incubated in E3 until 50 hpf, when remaining rhodamine-dextran was detected by confocal microscopy.

## Sectioning of embryos

Cryosections were generated as previously described (*Westcot et al., 2015*). For TEM analysis, embryos were anesthetized with Tricaine, heads were removed using a scalpel and subjected to genotyping, and tails were processed for ultrathin sectioning and TEM as described (*Feitosa et al., 2011*).

## Immunofluorescence (IF) analyses

To determine cell proliferation, embryos were incubated in 10mM BrdU in E3 for 1 or 2 hr, followed by a one-hour wash with E3 and fixation in 4% PFA. BrdU incorporation was detected by anti-BrdU immunolabelling. IF analyses were performed essentially as previously described (*Carney et al., 2007*). Embryos were fixed in 4% PFA for stainings using the following primary antibodies: mouse anti-Tjp1 (Zymed, San Francisco, CA; 33–9100, 1:200), mouse anti-chicken ATPa5 (Developmental Studies Hybridoma Bank; DSHB, 1:200), mouse anti-chicken ATPa6F (DSHB, 1:200), rabbit anti-Cdh1 (Anaspec, Fremont, CA; 1:200), mouse anti-Cdh1 (BD Biosciences, 610188, 1:200), rabbit anti-laminin (Sigma Aldrich, St. Louis, MO; L9393, 1:200), mouse anti-BrdU (Roche, Basel, CH; 1170376, 1:100), mouse anti-collagen II (DSHB II-II6B3-c, 1:200), chicken anti-GFP (Invitrogen; A10262, 1:300), mouse anti-p63 BC4A4 (Zytomed, 1:200), rabbit anti-zebrafish Lgl2 (*Sonawane et al., 2009*) (1:400), rabbit anti-aPKC (C-20, Santa Cruz Biotechnologies, Dallas, TX; sc-216, 1:200), rabbit anti-phospho-S6 ribosomal protein (pS6RP, Ser240/241; Cell Signaling Technology, Danvers, MA; #2215, 1:300). For stainings with the mouse anti-panKeratin1-8 (Progen Pharmaceuticals, Darra, Australia; 61006, 1:10), embryos were fixed with Dent's fixative (80% Methanol, 20% DMSO, at -20°C overnight). For pAkt stainings, embryos were fixed in EAF (40% ethanol, 5% acetic acid, 4% formaldehyde in PBS) and either washed with PBS-TritonX, followed by an antigen retrieval in 10 mM Tris, 1 mM EDTA, pH 9.0 for 60 min at 60°C, blocking in 5% sheep serum and antibody incubation with rabbit anti-pAkt (S473) (Cell Signaling Technology #4060, 1:50) or processed for cryosectioning as described (*Westcot et al., 2015*). Secondary antibodies were anti-mouseCy3, anti-rabbitCy3, anti-mouseAlexa488, anti-mouseAlexa647, and anti-chickenAlexa488 (all Invitrogen, Carlsbad, CA; 1:200). Images were taken using a Zeiss Confocal (LSM710 META) and processed using the ImageJ software.

## Cell transplantations

Ventral ectodermal cells from *Tg(Ola.Actb:Hsa.hras-egfp)*[vu119] donor embryos either un-injected or injected with *atp1b1a* MO were transplanted into the ventral ectoderm of wild-type or *atp1b1a* morphant recipients at 6 hpf. Homotypic mosaics (wt to wt, morphant to morphant) embryos were raised in E3 medium until 48 hpf, mounted in 1.5% LMP agarose in E3, and clusters of GFP+ cells were recorded by time-lapse confocal microscopy (*Figure 2A,B* and *Videos 1,2*). Heterotypic chimeras (wt to morphant, morphant to wt) were raised in E3 250 mM mannitol until 84 hpf, fixed in Dent's fixative and subjected to anti-GFP and cytokeratin IF (*Figure 8a,b*).

**Table 1.** Sequences and concentrations of antisense morpholino oligonucleotides (MOs) used.

| MO | Sequence | Concentration | Ref. |
|---|---|---|---|
| pu.1 | GATATACTGATACTCCATTGGTGGT | 0.8 mM | (*Carney et al., 2007*) |
| atp1b1a | CGGTATTTAGTTCCCTTTTTGGTGG | 0.75 mM (full), 0.01 mM (subphenotypic) | (*Blasiole et al., 2006*) |
| lgl2 | GCCCATGACGCCTGAACCTCTTCAT | 0.05 mM | (*Sonawane et al., 2005*) |
| mmp9 | CGCCAGGACTCCAAGTCTCATTTTG | 0.2 mM | (*LeBert et al., 2015*) |

## Inhibitor treatments

Ouabain (Sigma-Aldrich; St. Louis, MO) and hydroxyurea (Merck, Kenilworth, NJ) were dissolved in water, Wortmannin (Sigma-Aldrich), PIK90 (Merck), LY294002 (TocrisBioscience, Bristol, UK), Rapamycin (Merck), AZD8055 (Santa Cruz Biotechnologies), and Withaferin A (TocrisBioscience) in DMSO, and solutions were further diluted in E3 embryo medium to concentrations of 3mM (ouabain), 50mM (hydroxyurea), 1 µM (Wortmannin), 5 µM (PIK90), 25 µM (LY294002), 1.1 µM (Rapamycin), 30 µM (AZD8055), and 30 µM (WithaferinA). Embryos were incubated in inhibitor solution starting from 34 hpf and scored at 54 hpf.

## Morpholino knockdown

The following morpholinos were obtained from GeneTools (Philomath, CA) and 1.5 nl injected in 1-cell stage embryos according to standard protocols:

## Statistics

Quantitative experiments were repeated at least three times, reaching similar results. No outliers were encountered. Mean values and standard deviations of all individual specimens (biological samples; n) from one representative or all independent experiments are presented, as specified in the respective figure legends. Numbers of biological samples analyzed were decided on depending on obtained standard deviations and statistical significances. Quantifications of phenotypes were conducted by comparing treated and control progeny of the same parental fish. Fluorescence intensities were determined using ImageJ software. Significance of differences was determined using an unpaired two-tailed Student's t-test, and obtained p values are mentioned in the respective figure legends. Numerical data from all quantitative analyses are also provided in the Supplement as a source data file. Data illustrated in representative images that are shown without statistical calculations were obtained from at least 15 out of 15 investigated specimens from at least three independent experiments.

## Acknowledgements

We thank Heike Wessendorf for excellent technical assistance; David Kimelman, Gilbert Weidinger, Steve Renshaw, Zhiyuan Gong and Jacek Topczewski for zebrafish lines; and Mahendra Sonawane for the Lgl2 antibody. Work in the laboratory of MH was supported by the German Research Foundation (DFG; SFB 829 and its Z2 project), the European Union (Seventh Framework Programme, Integrated Project ZF-HEALTH, EC Grant Agreement HEALTH-F4-2010-242048) and the US National Institute of General Medical Sciences (GM63904).

## Additional information

### Funding

| Funder | Grant reference number | Author |
|---|---|---|
| Deutsche Forschungsgemeinschaft | SFB 829 | Matthias Hammerschmidt |
| National Institute of General Medical Sciences | GM63904 | Matthias Hammerschmidt |

| European Commission | HEALTH-F4-2010-242048 | Matthias Hammerschmidt |

The funders had no role in study design, data collection and interpretation, or the decision to submit the work for publication.

## Author contributions

JH, Conceived, Designed and performed all zebrafish experiments, Analyzed and interpreted the data, Wrote the manuscript with input from all authors, Conception and design, Acquisition of data, Analysis and interpretation of data, Drafting or revising the article; FB, BW, Performed whole genome sequencing and analyzed and interpreted sequences, Acquisition of data, Analysis and interpretation of data; HH, WB, Performed electron microscopy and analyzed and interpreted micrographs, Performed TEM analysis, Acquisition of data, Analysis and interpretation of data; JA, PN, Performed whole exome sequencing and sequence data analysis, Acquisition of data; MH, Conceived the project, Designed the study, Analyzed and interpreted data, Wrote the manuscript with input from all authors, Conception and design, Analysis and interpretation of data, Drafting or revising the article

## Author ORCIDs

Matthias Hammerschmidt, http://orcid.org/0000-0002-3709-8166

## Ethics

Animal experimentation: All zebrafish experiments were approved by the national animal care committees (LANUV Nordrhein-Westfalen; 8.87-50.10.31.08.129; 84-02.04.2012.A251; City of Cologne; 576.1.36.6.3.01.10 Be) and the University of Cologne.

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
