## [Decision Letter]

Thank you for submitting your work entitled "Tumor suppression in basal keratinocytes via dual non-cell-autonomous functions of a Na,K-ATPase β subunit" for consideration by *eLife*. Your article has been favorably evaluated by Fiona Watt (Senior editor) and three reviewers, one of whom is a member of our Board of Reviewing Editors.

The reviewers have discussed the reviews with one another and the Reviewing Editor has drafted this decision to help you prepare a revised submission.

Summary:

This paper deals with the study of uncontrolled basal cell proliferation in zebrafish, as a model to investigate carcinogenesis and tumor suppressors. The major conceptual advance of this paper is the claim that systemic isotonicity prevents malignant transformation, thereby identifying hypotonic stress as a new contributor to tumor development.

Essential revisions:

Both points from reviewer 1.

Points #2 (using MO's for example) and #6 from reviewer 2.

Addressing point #1 from reviewer 2 (through mosaic analysis for example) would be a terrific addition to the paper, and points #2 and 5 can be taken care of by appropriate changes in the text.

Reviewer #1:

This paper from the Hammerschmidt lab deals with the study of uncontrolled basal cell proliferation in zebrafish, as a model to investigate carcinogenesis and tumor suppressors. The major conceptual advance of this interesting paper is the claim that systemic isotonicity prevents malignant transformation, thereby identifying hypotonic stress as a new contributor to tumor development.

The data are generally convincing and of high quality. I have a few concerns:

1) The evidence that the psoriasis phenotype is caused by a loss-of-function mutation in the *atp1b1a* gene is fairly convincing but the following data should be added to the supplementary information: map showing 10-20 genes on each side of *atp1b1a* as well as a list of any other genes in this region that were sequenced to identify molecular lesions.

2) In order to tighten the text, it might make sense to move the negative data to the supplementary material (e.g., panels 10c-f; panels 5b-c could also be moved). As it stands, the paper is very long winded.

Reviewer #2:

The manuscript entitled "Tumor suppression in basal keratinocytes via dual non-cell-autonomous functions of a Na,K-ATPase β subunit" by Hatzold et al. reports the importance of *atp1b1a* function in prevention of malignancy in the zebrafish epidermis. The authors propose that perturbed cell adhesion, cell polarity in the epidermis and osmotic stress caused due to impaired function of heart and larval kidney contribute towards the development of malignancy. They further show that the osmotic stress results in activation of PI3K-AKT-mTORC1- NFκB-MMP9 pathway in the basal epidermis promoting cell proliferation as well as invasive behavior of the cells. These findings are conceptually interesting and bring into light the importance of osmotic stress in development of malignancy. Following concerns need to be addressed to improve the analyses and to substantiate the authors' claims.

1) EMT is one of the major hallmarks of Malignancy. The authors should perform live imaging of the basal cells to show EMT in the psoriasis mutant.

2) The basic premise of this paper is that the loss of cell polarity and adhesion in the periderm secondarily affect the adhesion and polarity in the basal epidermis in psoriasis mutant. These perturbations in cell polarity and adhesion in conjunction with the osmotic stress results in malignant behavior of the basal epidermal cells. If correct, driving expression of *atp1b1a* in the periderm should rescue all the aspects of cell polarity phenotype in the periderm as well as in the basal epidermis and also rescue malignancy in the basal epidermis without affecting the edema phenotype (as the kidney and heart will remain nonfunctional) in hypotonic medium. Authors have just looked at the keratin localization upon driving the expression of *atp1b1a* in the periderm, that too in the isotonic condition, which is not enough. These rescue experiments are crucial for this paper and can be easily done using their transgenic line that drives expression of *atp1b1a-gfp* in periderm under keratin4 promoter.

3) Authors suggest that the decrease in heart-beats and lowering the blood pressure affects kidney function and leads to edema formation. This is far-fetched in the absence of any solid evidence. It is equally possible that the effect on polarity of kidney epithelial cells results in edema development, which further decreases the heart function.

4) There is a considerable decrease in Lgl2 localisation in the psoriasis mutant epidermis. This result indicates that Lgl2 localisation requires *atp1b1a* function in the periderm (possibly directly) as well as in the basal epidermis (indirectly?) whereas the interaction studies indicate that *atp1b1a* and *lgl2* have partial redundant functions. Previous studies (Reischauer et al., Westcot et al.) suggest that Lgl has a pro basal function in the basal cells of the developing fin epithelium and that it's loss is enough to cause hyperplasia and EMT. Therefore, it is difficult to interpret the genetic interaction data that authors have. Is it possible that the decrease in Lgl levels contributes to the malignancy phenotype in psoriasis mutant? If yes, can authors rescue the aspects of malignancy by restoring the Lgl levels in psoriasis mutants? Is it that the reduction in Lgl and ATP1b1a levels in the periderm results in malignant phenotypes when morpholinos are coinjected at suboptimal concentrations? Or is it that the reduction in Lgl levels in the basal epidermis and reduction of Atp1b1a in the periderm results in the malignant phenotype when authors co-inject the morpholinos at sub optimal concentrations? What is the status of edema in the larvae injected with the suboptimal concentration of both the morpholinos? Is it the same PI3K-AKT-mTORC1- NFκB-MMP9 pathway that gets activated in the co-injected embryos? In its current form the genetic interaction data does not contribute much towards our understanding of how these two components actually interact and function together in stratified epithelium. If authors wish to include these data in the manuscript a thorough analysis of this part (addressing questions raised above) is required.

5) The malignant phenotypes are mainly observed in the fins. Are the polarity phenotypes present throughout the epidermis? If yes, how do the authors explain the prevalence of the malignant phenotype in the fin? The expression analysis indicates that *atp1b1a* is expressed mostly in the flank and in the fins. Is this the reason for stronger phenotypes in the fin? In general it will be good if authors mention in the main text where the localization analysis is performed for polarity or adhesion components.

6) The rescue experiments using inhibitors of signaling pathway components (NFkB, mTOR etc) should be accompanied by BrdU analysis. Since authors argue that these components act in a linear pathway, it is essential to show that inhibition of PI3K restores the pAKT levels in the mutant whereas treatment with Rapamycin and NFkB inhibitor does not reduce the pAKT levels. In addition, a readout for mTOR signaling (phospho S6K) can be used to place mTOR in between PI3K and NFkB.

Reviewer #2 (Additional data files and statistical comments):

Since the effect of rapamycin treatment is weaker as compared to the other inhibitors, the number of experimental animals for rapamycin treatment may be increased. The number (n=31) is low, possibly with 9 mutant animals of which 4 do show weak aggregates in the mutant. It would be useful if authors describe the criterion for weak and strong phenotypes.

Reviewer #3:

This work describes the very interesting finding that mutation of a Na,K-ATPase regulatory subunit causes in vivo epidermal malignancy by a novel mechanism. The work starts out with a very interesting zebrafish mutant called psoriasis that had previously not been genetically identified (despite much effort), but is identified here with the advent of whole exome sequencing, and found, surprisingly to be the protein mentioned above. This is surprising since one of the hallmarks of psoriasis is the overgrowth of the epidermal cells and an ATPase would not seem an obvious candidate to be a tumor suppressor.

With the key gene identified, the authors then go on to do a set of very interesting experiments, including showing the very unexpected result that changing the hypotonicity of the growth media inhibits the epidermal growths, revealing a non-autonomous osmoregulatory role of the Na,K-ATPase in these tumors. In addition, they also show that the overgrowth effects on the basal epidermal cells is due to effects in the overlying periderm, again showing a surprising non-autonomous effect. Finally, they also provide molecular details, connecting these changes to the Akt/NFkB pathway.

This work provides some very interesting findings and shows very nicely how the fish system can be used to figure out complex oncological pathways, and to discover how they are regulated in the whole animal, not just within the tumor per se. This will be of interest to a wide range of readers. There is a large amount of work, and the writing is very clear. I fully support publication.

---

## [Author Response]

Essential revisions:

*Both points from reviewer 1.*

*Points #2 (using MO's for example) and #6 from reviewer 2.*

Addressing point #1 from reviewer 2 (through mosaic analysis for example) would be a terrific addition to the paper, and points #2 and 5 can be taken care of by appropriate changes in the text.

Points #1 and #2 of Reviewer 1 have both been addressed as requested, the figures have been reorganized accordingly, and the requested additional information is now provided. In addition, we now provide all additional data requested in points #1, 2 and 6 of Reviewer 2. Furthermore, although not demanded by the Reviewing Editor, we have also carried out some of the experiments proposed by Reviewer 2 in point #4 and provide more data elucidating the cooperation between Atp1b1a and Lgl2. Furthermore, we have re-worded and extended our description and discussion of the role of Atp1b1a in heart and kidney, as requested by Reviewer 2 in point 3, and have clarified the topology of epidermal defects in atp1b1a mutants, as questioned by Reviewer 2 in point 5.

*Reviewer #1:*

1) The evidence that the psoriasis phenotype is caused by a loss-of-function mutation in the atp1b1a gene is fairly convincing but the following data should be added to the supplementary information: map showing 10-20 genes on each side of atp1b1a as well as a list of any other genes in this region that were sequenced to identify molecular lesions.

We now show the requested map as well as a table with 22/20 genes on each side of *atp1b1a*, and an indication of the interval to which *atp1b1a* had been meiotically mapped by Webb et al., 2008 (Figure 3—figure supplements 1 and 2). In the legends of these supplementary figures, as well as in the main text (subsection “The *psoriasis* phenotype is caused by a loss-of-function mutation in *atp1b1a* encoding a Na,K-ATPase β-subunit”, first paragraph), we also state that we sequenced the exons of all of these annotated genes. However, apart from the described non-sense mutation in *atp1b1a*, no further mutagenic lesions were identified.

2) In order to tighten the text, it might make sense to move the negative data to the supplementary material (e.g., panels 10c-f; panels 5b-c could also be moved). As it stands, the paper is very long winded.

We have removed the suggested panels to the Supplement, as well as multiple panels showing representative live images of embryos from additional figures, while just leaving the graphs with the quantifications in the main figures. In addition, to streamline the results on the PI3K-AKT-mTORC1-NFkB, which indeed were long winded, we re-organized former Figure 8 and Figure 9 to one, more systematically structured Figure, in which even more data were integrated, as requested by Reviewer #2 (new Figure 9).

*Reviewer #2:*

*1) EMT is one of the major hallmarks of Malignancy. The authors should perform live imaging of the basal cells to show EMT in the psoriasis mutant.*

Time-lapse live imaging of clones of mGFP-labeled basal kerationcytes of homotypically mosaic control and *atp1b1a* morphant embryos were generated, starting at 48 hpf and outside epidermal aggregates. Two representative videos are now supplied as supplemental material, and stills of these two videos are shown as new Figure 2. In comparison to wild-type cells, which are very static, *atp1b1a* morphant cells are much more dynamic, with protrusive activity and ongoing dissociation and re-association with neighboring cells. These results are now described in the second paragraph of the subsection “*psoriasis* mutant embryos display characteristics of epidermal malignancy”. We name this behavior a “partial EMT”, similar to what we have formerly observed for morphants lacking the Matriptase inhibitor Hai1a (Carney et al., 2007).

*2) The basic premise of this paper is that the loss of cell polarity and adhesion in the periderm secondarily affect the adhesion and polarity in the basal epidermis in psoriasis mutant. These perturbations in cell polarity and adhesion in conjunction with the osmotic stress results in malignant behavior of the basal epidermal cells. If correct, driving expression of atp1b1a in the periderm should rescue all the aspects of cell polarity phenotype in the periderm as well as in the basal epidermis and also rescue malignancy in the basal epidermis without affecting the edema phenotype (as the kidney and heart will remain nonfunctional) in hypotonic medium. Authors have just looked at the keratin localization upon driving the expression of atp1b1a in the periderm, that too in the isotonic condition, which is not enough. These rescue experiments are crucial for this paper and can be easily done using their transgenic line that drives expression of atp1b1a-gfp in periderm under keratin4 promoter.*

The reason for having performed the experiments in isotonic medium was to eliminate any effects possibly caused by the systemic hypotonicity, rather than the loss of *atp1b1a* in the periderm. Nevertheless, we have now also incorporated the requested studies of *Tg(krt4:atp1b1a-gfp)–*containing mutants kept in hypotonic medium (additions to Figure 8 and new Figure 8—figure supplement 1). Both for epidermal morphology/aggregate formation (Figure 8—figure supplement 1), as well as for the malignancy markers pAKT (Figure 8) and *mmp9* (Figure 8—figure supplement 1), we obtained a significant, yet incomplete restoration to wild-type conditions, whereas edema remained unaffected (see Figure 8—figure supplement 1 for the quantification of epidermal aggregate and edema formation; in addition, for the different stainings, results similar to the shown representatives were obtained for at least 15 out of 15 investigated specimen from at least three independent experiments; subsection “Statistics”). Of note, a similar partial restoration as for the malignancy markers was obtained for the cell polarity markers Lgl2 in the periderm (Figure 8—figure supplement 1) and cytokeratin in the basal layer (Figure 8—figure supplement 1). In conclusion, while further supporting the notion that Atp1b1a from the periderm promotes epithelial polarity and suppresses malignancy in basal keratinocytes, the data further indicate that the degrees of polarity defects and epidermal malignancy are proportionally linked. In addition, they suggest that hypotonic stress further challenges the epithelial cell polarity system, which therefore is more difficult to repair by transgene-encoded Atp1b1a from the periderm than under isotonic conditions. In the text of the manuscript, these new data are described and discussed in the last paragraph of the subsection “*atp1b1a* interacts with *lgl2* and is required in peridermal cells to establish proper epithelial organization of basal keratinocytes”.

3) Authors suggest that the decrease in heart-beats and lowering the blood pressure affects kidney function and leads to edema formation. This is far-fetched in the absence of any solid evidence. It is equally possible that the effect on polarity of kidney epithelial cells results in edema development, which further decreases the heart function.

We never meant to claim this and actually don’t think we did. But, we have re-written the corresponding paragraphs in the Results (subsection “*atp1b1a* is required for proper heart and pronephric function”) and Discussion (subsection “The role of Atp1b1a in kidney and heart to avoid systemic hypotonicty”) to point out more clearly the direct effects of loss of Atp1b1a function in pronephric epithelial cells as the likely primary cause of aberrant osmoregulation. In the Discussion, we also mention that in addition to the mislocalization of the pump’s α subunit, epithelial polarity of renal epithelial cells might be generally compromised.

*4) There is a considerable decrease in Lgl2 localisation in the psoriasis mutant epidermis. This result indicates that Lgl2 localisation requires atp1b1a function in the periderm (possibly directly) as well as in the basal epidermis (indirectly?) whereas the interaction studies indicate that atp1b1a and lgl2 have partial redundant functions. Previous studies (Reischauer et al., Westcot et al.) suggest that Lgl has a pro basal function in the basal cells of the developing fin epithelium and that it's loss is enough to cause hyperplasia and EMT. Therefore, it is difficult to interpret the genetic interaction data that authors have. Is it possible that the decrease in Lgl levels contributes to the malignancy phenotype in psoriasis mutant? If yes, can authors rescue the aspects of malignancy by restoring the Lgl levels in psoriasis mutants? Is it that the reduction in Lgl and ATP1b1a levels in the periderm results in malignant phenotypes when morpholinos are coinjected at suboptimal concentrations? Or is it that the reduction in Lgl levels in the basal epidermis and reduction of Atp1b1a in the periderm results in the malignant phenotype when authors co-inject the morpholinos at sub optimal concentrations? What is the status of edema in the larvae injected with the suboptimal concentration of both the morpholinos? Is it the same PI3K-AKT-mTORC1- NFκB-MMP9 pathway that gets activated in the co-injected embryos? In its current form the genetic interaction data does not contribute much towards our understanding of how these two components actually interact and function together in stratified epithelium. If authors wish to include these data in the manuscript a thorough analysis of this part (addressing questions raised above) is required.*

In general, we agree with the Reviewing Editor (who did not mark this point as an essential revision) that these proposed experiments, although interesting, are beyond the scope of this manuscript, as its main focus is on the tumor-suppressive role of ATP1b1a, and not on it’s the mechanisms underlying its cooperation with Lgl2. The main reason to show the genetic interaction between the two was the desire to provide a further line of evidence that ATP1b1a regulates epithelial cell polarity of epidermal cells. Therefore, we very much prefer to leave these interaction data in the manuscript, in particular, as we now have added at least some (in our opinion the most crucial) of the proposed new experiments. Thus, we now show that embryos co-injected with sub-phenotypic doses of *atp1b1a* and *lgl2* MOs also display edema formation and a down-regulation of Na/K-ATPase α-subunits in basolateral domains of pronephric epithelial cells, as well as an up-regulation of pAKT levels and *mmp9* expression in basal keratinocytes. Moreover, we show that the epidermal aggregates, but not the edema, are rescued upon treatment with the chemical PI3K, mTORC1 and NFkB inhibitors (new Figure 7—figure supplement 1). We conclude that ATP1b1a and Lgl2 most likely cooperate both in the epidermis (periderm) and in the pronephros to suppress epidermal malignancy. These new findings are now described and discussed in the first paragraph of the subsection “*atp1b1a* interacts with lgl2 and is required in peridermal cells to establish proper epithelial organization of basal keratinocytes”, and in the subsection “The cooperation between Atp1b1a and Lgl2”.

*5) The malignant phenotypes are mainly observed in the fins. Are the polarity phenotypes present throughout the epidermis? If yes, how do the authors explain the prevalence of the malignant phenotype in the fin? The expression analysis indicates that atp1b1a is expressed mostly in the flank and in the fins. Is this the reason for stronger phenotypes in the fin? In general it will be good if authors mention in the main text where the localization analysis is performed for polarity or adhesion components.*

We now show that the malignant phenotypes (epidermal aggregates) also occur on the flank, the yolk sac, and the head of the mutant embryos, so basically everywhere (new Figure 1). Generally, for our molecular analyses of polarity and malignancy markers, we tried to avoid regions within epidermal aggregates, as the altered spatial organization of the epidermis in these aggregates makes it very difficult to image the molecular alterations at higher magnifications. However, the same molecular alterations were seen within aggregates and in inter-aggregate regions, independent of their position on median fins, flank, yolk sac or head. Anyway, we now state in the figure legends where the images were taken.

*6) The rescue experiments using inhibitors of signaling pathway components (NFkB, mTOR etc) should be accompanied by BrdU analysis.*

BrdU incorporation analyses have been added to the more systematically structured new Figure 9 (panels K-K’’’’).

*Since authors argue that these components act in a linear pathway, it is essential to show that inhibition of PI3K restores the pAKT levels in the mutant whereas treatment with Rapamycin and NFkB inhibitor does not reduce the pAKT levels.*

We had already shown in the last version of the manuscript that pAKT levels were restored by wortmannin (reduced), whereas they remained high after Whitaferin-A treatment. We have now added the corresponding data for Rapamycin-treated mutants (pAKT levels remain high) and show them next to each other in the newly structured Figure 9 (panels G-G’’’’).

In addition, a readout for mTOR signaling (phospho S6K) can be used to place mTOR in between PI3K and NFkB.

We now show immunofluorescence analyses with a pS6RP antibody formerly used in zebrafish as a readout of mTORC1 signaling, demonstrating that it is strongly up-regulated in psoriasis mutants compared to wild-type siblings, and down-regulated again to wild-type levels upon treatment with Wortmannin and Rapamycin, but not upon treatment with Withaferin A, in line with the proposed linearity of the pathway (Figure 9).